# Establishing a mass spectrometry-based system for rapid detection of SARS-CoV-2 in large clinical sample cohorts

Karina Helena Morais Cardozo [1,3], Adriana Lebkuchen [1,3], Guilherme Gonçalves Okai [1,3], Rodrigo Andrade Schuch [1,3], Luciana Godoy Viana [1,3], Aline Nogueira Olive [1], Carolina dos Santos Lazari[2], Ana Maria Fraga [1], Celso Francisco Hernandes Granato [2], Maria Carolina Tostes Pintão [1] & Valdemir Melechco Carvalho [1]✉

The outbreak of severe acute respiratory syndrome coronavirus 2 (SARS-CoV-2) is pressing public health systems around the world, and large population testing is a key step to control this pandemic disease. Here, we develop a high-throughput targeted proteomics assay to detect SARS-CoV-2 nucleoprotein peptides directly from nasopharyngeal and oropharyngeal swabs. A modified magnetic particle-based proteomics approach implemented on a robotic liquid handler enables fully automated preparation of 96 samples within 4 hours. A TFC-MS system allows multiplexed analysis of 4 samples within 10 min, enabling the processing of more than 500 samples per day. We validate this method qualitatively (Tier 3) and quantitatively (Tier 1) using 985 specimens previously analyzed by real-time RT-PCR, and detect up to 84% of the positive cases with up to 97% specificity. The presented strategy has high sample stability and should be considered as an option for SARS-CoV-2 testing in large populations.

[1] Division of Research and Development, Fleury Group, 04344-070 São Paulo, SP, Brazil. [2] Division of Infectious Diseases, Fleury Group, 04344-070 São Paulo, SP, Brazil. [3] These authors contributed equally: Karina Helena Morais Cardozo, Adriana Lebkuchen, Guilherme Gonçalves Okai, Rodrigo Andrade Schuch, Luciana Godoy Viana. ✉email: valdemir.carvalho@grupofleury.com.br

The outbreak of a novel coronavirus (SARS-CoV-2) was first identified at the end of 2019, and is challenging the health systems worldwide[1]. The disease has rapidly spread around the world with over 24 million reported cases and more than 820,000 deaths confirmed as of 31 August 2020[2].

Tests using real-time reverse transcription-polymerase chain reaction (real-time RT-PCR) were quickly developed after the viral genome sequence release[3]. Although highly sensitive, the soaring demand for this test caused a shortage of several reagents and instrumentation used for this method, severely limiting its applicability to large-scale screening[4,5].

Point-of-care tests are the most desirable alternative to ramp up large-scale population screening. Currently, hundreds of initiatives are on course to deliver serologic and antigen detection platforms. Serologic immunoassays targeting IgA, IgM, or IgG provide historic information about viral exposure, but their sensitivity in the acute phase of SARS-CoV-2 infections is still not well established[6]. At the time of publishing, immunoassays directed towards antigen detection are being delivered but scarce information about their sensitivity and specificity is available[7–9]. Immunoassays can be highly sensitive, simple to perform, and provide quick answers at a reasonable cost. However, they often suffer from interference because antibody recognition is not free of error and may be confounded by the presence of other molecules.

Mass spectrometry (MS) has become an essential tool in clinical laboratories and is the current gold standard for several clinical applications such as steroid hormone determination[10]. Matrix-assisted laser desorption/ionization time-of-flight mass spectrometry (MALDI-MS) has been widely used to identify microbial species in clinical samples[11,12]. Recently, a diagnostic method based on MALDI-MS for direct SARS-CoV-2 detection from nasal swabs with high accuracy was described[13]. This strategy relied on spectral features selected by machine learning, but the chemical identities of the measured markers remain unknown.

Targeted proteomics, a derivative from proteomic-based MS technology, has emerged as a viable alternative to immunoassays for protein analysis[14]. Targeted proteomics methods have been applied to the routine determination of clinical biomarkers such as thyroglobulin[15], troponin I, myoglobin, lactate dehydrogenase B[16], apolipoproteins and glycated hemoglobin[17]. Generating targeted proteomics assays is faster than producing new antibodies and because MS-based assays are intrinsically more specific than immunoassays, this technique can be an interesting addition to viral testing panels.

Applications of targeted proteomics in the diagnosis of infectious diseases were limited until recently. Karlsson and colleagues demonstrated the use of nano-liquid chromatography coupled with tandem MS to identify four bacterial pathogens in respiratory tract samples (nasopharyngeal and nasal swabs)[18]. A few studies have described virus detection from clinical specimens. Majchrzykiewicz-Koehorst et al.[19] detected influenza A H3N2 and H1N1 in highly pure and concentrated samples obtained by culturing viruses in cell lines and spiked throat swabs. In addition to the time required to expand viruses in cultured cells, their strategy required 3 h to analyze 10 samples. Proteomics-based detection of SARS-CoV-2 has been recently described after virus expansion in cell culture detection[20,21].

Foster and colleagues[22] pioneered direct virus detection from clinical samples by MS-based targeted proteomics. Human metapneumovirus (HMPV) was detected directly from nasopharyngeal aspirates after concentration via size-exclusion chromatography, overnight trypsin digestion, and detection using 30-min multiple reaction monitoring assay. Recently, several research groups have proposed strategies based on targeted proteomics to detect SARS-CoV-2 in clinical specimens[23–26]. However, lengthy and laborious manual sample preparation procedures[23–25] and

long analysis times for liquid chromatography separation and MS detection[24,26] would limit their application in large-scale testing. Besides, their analytical performance was not fully demonstrated since they relied on preliminary validations with a limited number of clinical samples[24–26].

Here, we set out to develop a rapid, specific, and robust method to enable high-throughput screening to support large-scale SARS-CoV-2 clinical diagnostics. For this purpose, two different assays are validated based on the parameters for MS-based targeted proteomics assays: qualitative (Tier 3) and quantitative (Tier 1)[27]. We first analyze clinical respiratory tract samples using a bottom-up proteomics workflow, resulting in a spectral library that we use to generate a list of targeted peptides. The peptide selection is further refined using parallel reaction monitoring (PRM) on a microflow chromatography–high-resolution MS platform. The selected SARS-CoV-2 peptides are then used to develop a high-throughput targeted proteomics assay based on turbulent flow chromatography coupled to tandem mass spectrometry (TFC-MS/MS). This proteomics assay does not detect peptides in clinical specimens from other coronavirus strains, rhinovirus, enterovirus, and influenza viruses, suggesting that there is no assay interference. When applied to several hundred clinical samples, the assay detects up to 84% of the SARS-CoV-2 positive cases identified by an in-house real-time PCR method, demonstrating the utility of this MS-based approach for high-volume SARS-CoV-2 testing.

## Results

**Selection of target peptides by untargeted analysis.** Respiratory tract specimens previously analyzed by an in-house real-time RT-PCR method were directly processed by a shotgun proteomics protocol to investigate the presence of predicted peptides from the UniProt SARS-CoV-2 pre-release (downloaded on 13 March 2020). To increase the probability of detection of SARS-CoV-2 proteins, samples with low-cycle threshold values (Ct), which correspond to high viral load, were selected. Simple ethanol precipitation was used to concentrate proteins from nasopharyngeal/oropharyngeal swab specimens conserved in virus transport medium. Protein pellets were lysed by sodium dodecyl sulfate (SDS), reduced, and digested by trypsin. Data-dependent acquisition (DDA) analyses revealed the presence of 119 unique peptides from eight proteins out of the 14 proteins predicted by UniProt SARS-CoV-2 (Table 1). Nucleoprotein (NCAP_WCPV) accounted for 23.5% of the identified peptides and had a sequence coverage of 72.3%. The DDA experiments were used to create a spectral library for PRM design (Fig. 1). After filtering out missed cleavages and cysteine-containing peptides, 17 peptides were selected: nine from the nucleoprotein, five from the spike glycoprotein, two from the membrane protein, and one from the protein 3a (Supplementary Fig. 1). Using a first-round 60-min PRM acquisition, nucleoprotein peptides were found to exhibit ~80-fold higher relative intensities compared to peptides from other proteins and were thus selected for SARS-CoV-2 detection. A fast PRM method was achieved with a 9-min microflow chromatography separation using eight targeted peptides from nucleoprotein protein (Supplementary Fig. 2).

The specificity of the targeted peptides was also confirmed by blastp against the UniProt database. The presence of mutations in the targeted peptides was excluded after inspecting 18 individual nucleoprotein sequences that included all local sequences deposited in GISAID (Supplementary Fig. 3). Notwithstanding, the evaluation of 7666 genomes through SARS-CoV-2 Alignment Screen[28] detected the presence of mutations in the DGIIWVAT EGALNTPK and IGMEVTPSGTWLTYTGAIK coding regions (Supplementary Fig. 4 and Supplementary Table 1).

**Table 1 SARS-CoV-2 proteins identified by data-dependent analysis (DDA).**

| Protein group | Entry name | Protein name | Unique peptides | Sequence coverage (%) | Mol. weight (kDa) |
|---|---|---|---|---|---|
| P0DTC2 | SPIKE_WCPV | Spike glycoprotein | 22 | 24.4 | 141.18 |
| P0DTC3 | AP3A_WCPV | Protein 3a | 1 | 2.5 | 31.123 |
| P0DTC4 | VEMP_WCPV | Envelope small membrane protein | 2 | 33.3 | 83.649 |
| P0DTC5 | VME1_WCPV | Membrane protein | 4 | 20.3 | 25.146 |
| P0DTC7 | NS7A_WCPV | Protein 7a | 2 | 39.7 | 13.744 |
| P0DTC8 | NS8_WCPV | Non-structural protein 8 | 1 | 22.3 | 13.831 |
| P0DTC9 | NCAP_WCPV | Nucleoprotein | 28 | 71.1 | 45.625 |
| P0DTD1; P0DTC1 | R1AB_WCPV;R1A_WCPV | Replicase polyprotein 1ab; Replicase polyprotein 1a | 34;22 | 9.8 | 794.05 |

Proteins matched to SARS-CoV-2 (based on UniProt SARS-CoV-2 pre-release downloaded on 13 March 2020).
The following SARS-CoV-2 proteins were not detected: protein 7a, non-structural protein 7b, non-structural protein 8, ORF10 protein, protein 9b, and uncharacterized protein 14.

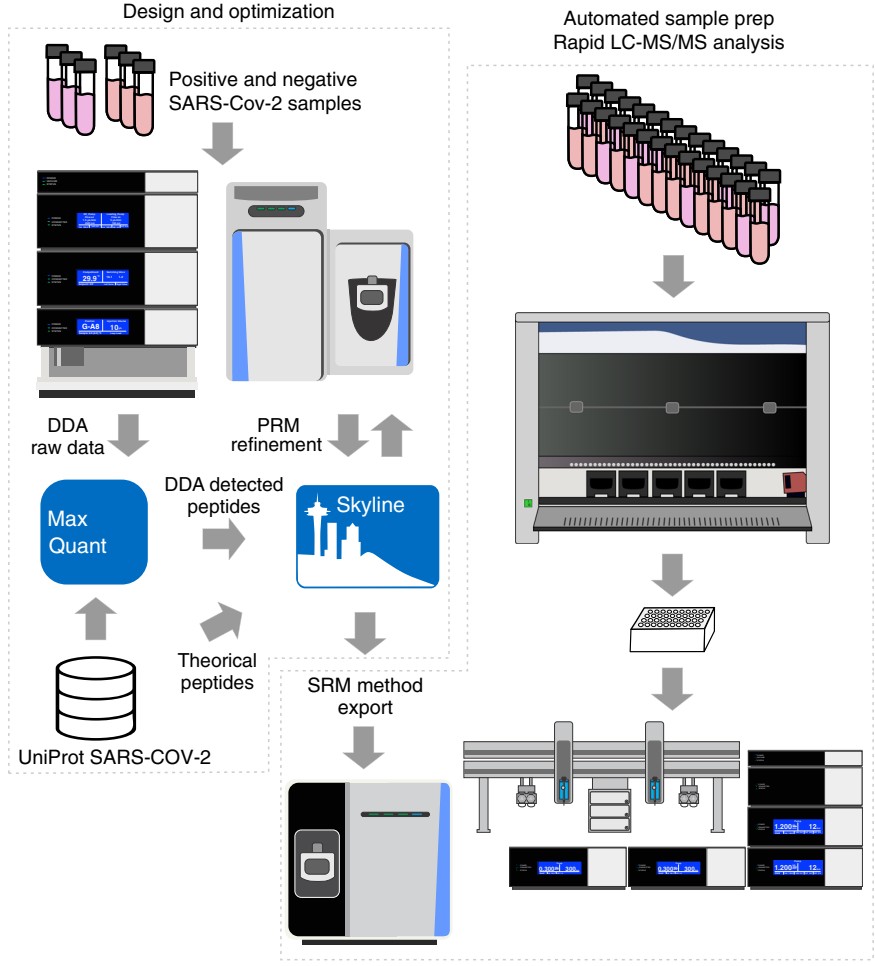

**Fig. 1 Test development workflow.** Design and optimization: peptide selection was performed by analyzing tryptic extracts from patient samples previously defined as positive or negative for SARS-CoV-2 by real-time RT-PCR. Data-dependent acquisitions (DDA) were achieved by microflow chromatography coupled to hybrid quadrupole-orbitrap tandem mass spectrometry. Skyline was used to generate the isolation list for parallel reaction monitoring (PRM). Several PRMs were performed before selection of the final two peptides that were exported as selected reaction monitoring (SRM) coordinates to a triple quadrupole tandem mass spectrometry. Automated sample prep/rapid LC-MS/MS (liquid chromatography coupled to tandem mass spectrometry) analysis: sample preparation was optimized, aiming for simplicity and speed and was fully implemented in a robotic liquid handler. Multiplexed four-channel analysis resulted in a 2.5-min acquisition time per sample.

**TFC coupled to MS detection**. To determine the best conditions for the analysis of targeted peptides by TFC, several conditions of loading, analyte transfer, washing, and re-equilibration were evaluated. Figure 2 illustrates the principle of turbulent flow and the multiplexing setup. For the loading step, different organic solvent contents (0, 5, and 10%) and formic acid concentrations (0, 0.1, and 0.5%) were tested of which the most efficient condition was 0.5% formic acid with no organic solvent content. The transfer step was the most critical, the usual elution by 100% organic solvent plug used in the focus mode did not yield

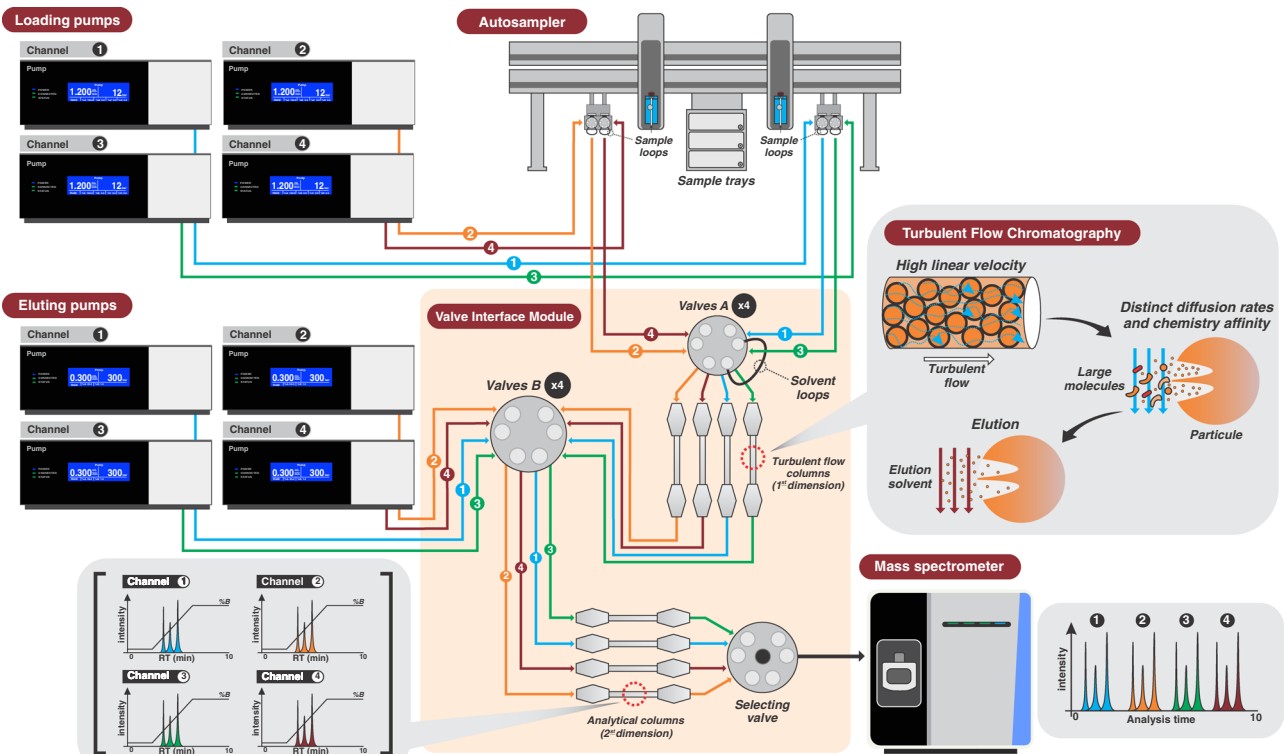

**Fig. 2 Schematic illustration of turbulent flow chromatography (TFC) setup in Transcend TLX-4 system coupled to TSQ Altis triple quadrupole mass spectrometer.** The TLX-4 system has four independent channels (channel 1: blue, channel 2: orange, channel 3: green and channel 4: red) that can be used simultaneously. The analytical run in each channel has three steps. The first step consists of sample injection, loading into the TFC column and cleanup. The sample is injected into the sample loop by autosampler and transferred to the valve interface module using a quaternary loading pump with high flow rate. For first dimension chromatography, the sample is loaded and washed into TFC column by a turbulent flow, flushing away unwanted matrix and large molecules while targeted analytes are retained in particles interaction sites. In the second step, the first and second columns are connected, and analytes are eluted from the TFC column by the solvent loop, transferred and focused on the analytical column head. Third step includes TFC column washing with stronger organic solvents, solvent loop refilling and ultraperformance chromatographic separation by an organic multistep linear gradient. As soon as the target analytes are eluted, the selecting valve directs the current channel flow path to the mass spectrometer (MS) to achieve data acquisition. Lastly, columns are re-equilibrated to initial conditions before starting the next run. For the present method, MS acquisition window has 2.5 min enabling multiplex analysis of up to four samples within ten minutes. Aria software coordinated multiplexed sequence logic to maximize analysis throughput.

good results. Different organic proportions were analyzed, and the best analyte transfer efficiency and peak shapes were achieved with 200 μL 60/40 0.5% formic acid/acetonitrile. The width of the transfer window was also evaluated in 6-s increments and 96 s provided the most efficient transfer for the selected peptides. It was observed that reducing flow in the loading pumps forward by 6 s before columns are in-line during the transfer step improved peak shape. A preliminary analysis indicated the presence of carryover after injection of samples with a high virus load. To investigate the source of carryover, the injection syringe was removed. The contamination persisted indicating it was not related to the syringe or injection port. Next, tubing from the injector to the valve interface module (VIM) was replaced and the carryover was still observed. Finally, changing the TFC column revealed it as the main source of contamination. Thus, several tests were performed to reduce TFC column contamination. Different organic solvents were essayed, including methanol, acetonitrile, isopropanol, acetone, dimethyl sulfoxide (DMSO), and trifluoroethanol (TFE). The incorporation of alternate flushing to TFC column with 20% DMSO/2% TFE in water followed by an organic solvent mixture (acetonitrile/isopropanol/acetone, 40:40:20, v/v) proved to be the most efficient in reducing the carryover. The concurrent analytical separation in the ultra-performance liquid chromatography (UPLC) column was achieved with a multistep linear

gradient. Finally, the elution loop was filled, the analytical column was flushed with 80% acetonitrile, and both columns were equilibrated for the following injection. Final chromatographic conditions as well as valve-switching programming are shown in Table 2.

The transition list for the selected peptides for SARS-CoV-2 was exported from the 9-min PRM Skyline method and imported into the TraceFinder Instrument Setup module. Among the initial eight target peptides there were important differences in polarity, two of the most hydrophobic peptides, namely DGIIWVATE-GALNTPK and IGMEVTPSGTWLTYTGAIK were more efficiently focused onto the analytical column. Although these two peptides were the ones less intense among those initially detected using the fast PRM method, they yielded sharp peaks and therefore better signal-to-noise ratios when employing the high-throughput method. The peptide HSGFEDELSEVLENQSS-QAELK from the fully [15]N-labeled chromogranin A used as a surrogate standard was included in the Tier 3 because it was detected within the chromatographic window of the SARS-CoV-2 peptides. A fully [15]N-labeled nucleoprotein was used as stable isotope-labeled (SIL) internal standard in the Tier 1. In addition, a peptide from endogenous beta actin (SYELPDGQVITIGNER), which eluted within the same chromatographic window, was also included in the Tier 1. Selected reaction monitoring (SRM) chromatograms for both tiers are shown in Fig. 3.

**Table 2 Conditions for turbulent flow chromatography (TFC) loading and eluting pumps for analysis of SARS-CoV-2 nucleoprotein target peptides.**

| Step | Start (min) | Loading pump | | | | | | | Eluting pump | | | | |
|---|---|---|---|---|---|---|---|---|---|---|---|---|---|
| | | Flow | Grad | %A | %B | %C | %D | Tee | Loop | Flow | Grad | %A | %B |
| 1 | 0 | 1.2 | Step | 100 | – | – | – | ==== | out | 0.3 | Step | 99 | 1 |
| 2 | 0.5 | 1.2 | Step | 100 | – | – | – | ==== | out | 0.3 | Step | 99 | 1 |
| 3 | 1 | 0.2 | Step | 100 | – | – | – | ==== | out | 0.3 | Step | 99 | 1 |
| 4 | 1.1 | 0.1 | Step | 100 | – | – | – | T | in | 0.3 | Step | 99 | 1 |
| 5 | 2.7 | 1.2 | Step | 60 | 40 | – | – | ==== | in | 0.3 | Step | 80 | 20 |
| 6 | 3.9 | 1.2 | Step | – | – | – | 100 | ==== | out | 0.3 | Ramp | 75.5 | 24.5 |
| 7 | 4.9 | 1.2 | Step | – | – | 100 | – | ==== | out | 0.3 | Ramp | 75.0 | 25 |
| 8 | 5.9 | 1.2 | Step | – | – | – | 100 | ==== | out | 0.3 | Ramp | 72.5 | 27.5 |
| 9 | 6.9 | 1.2 | Step | – | – | 100 | – | ==== | out | 0.3 | Ramp | 70 | 30 |
| 10 | 8.4 | 0.5 | Step | 100 | – | – | – | ==== | out | 0.3 | Ramp | 50 | 50 |
| 11 | 8.6 | 0.5 | Step | 100 | – | – | – | ==== | out | 0.3 | Step | 20 | 80 |
| 12 | 9.6 | 1.2 | Step | 100 | – | – | – | ==== | out | 0.3 | Step | 99 | 1 |

Steps 1 to 3: sample loading; step 4: transfer of peptides onto the analytical column; steps 5 to 10: switching wash between organic solvents (C and D) to clean large particles and reduce carryover in the TFC column, and peptide elution from the analytical column; step 11: TFC column equilibration and analytical column washing; step 12: system equilibration.
Loading: A (1–4) 0.5% formic acid in water; B (1–4) acetonitrile; C (1–4) isopropanol, acetonitrile, and acetone (40/40/20, v/v/v); D (1–4) 20% dimethyl sulfoxide (DMSO) and 2% trifluoroethanol in water.
Eluting: A (1–4) 1% DMSO, 0.1% formic acid in water; B (1–4) 1% DMSO, 0.1% formic acid in acetonitrile.

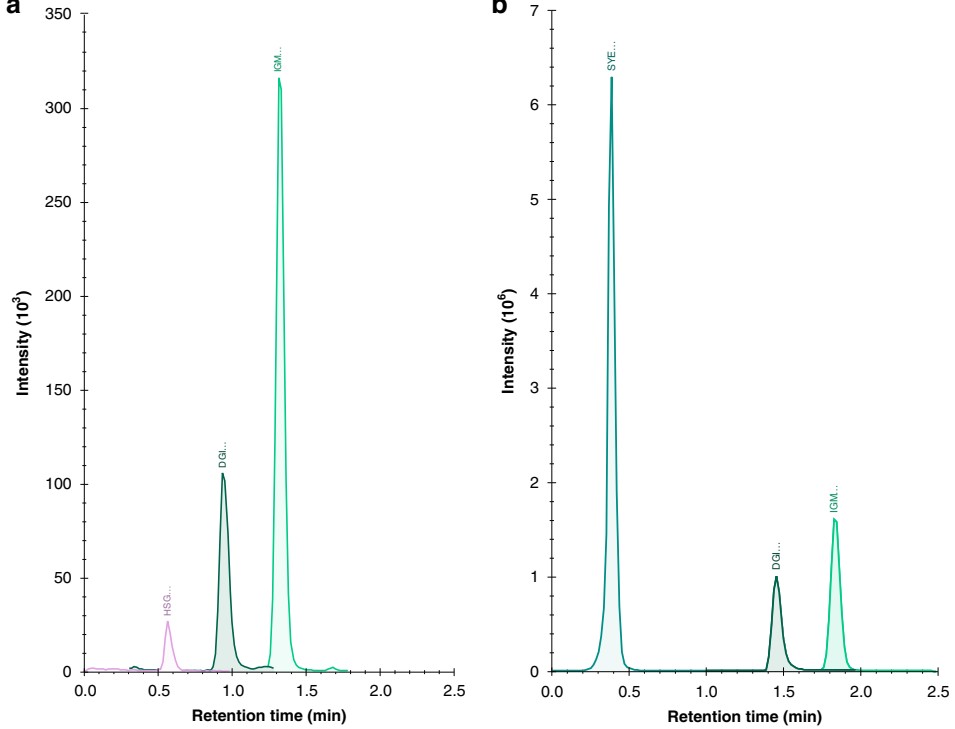

**Fig. 3 Selected reaction monitoring chromatograms of SARS-CoV-2-positive clinical respiratory tract specimen showing targeted peptides.**
**a** Qualitative (Tier 3) assay; **b** Quantitative (Tier 1) assay. The first three residues of each peptide are used to label peptide peaks: DGI (DGIIWVATEGALNTPK) and IGM (IGMEVTPSGTWLTYTGAIK) from nucleoprotein; HSG (HSGFEDELSEVLENQSSQAELK) from [15]N-labeled surrogate standard chromogranin A (in Tier 3) and SYE (SYELPDGQVITIGNER) from human beta actin (in Tier 1).

**Automated sample preparation.** From the initial nonautomated single-pot solid-phase-enhanced sample preparation (SP3) sample processing, several variables from bottom-up proteomics were investigated to reduce processing time of disulfide bonds reduction, alkylation, lysis, and digestion steps. The elimination of the alkylation step did not affect the detection of the target peptides. To assess the efficiency of protein capture on the magnetic beads, as well as other downstream steps such as digestion efficiency and LC-MS/MS detection, either a surrogate standard (fully [15]N-labeled chromogranin A) or SIL internal standard (fully [15]N-labeled nucleoprotein) were introduced in the first step of sample preparation. Just after the ethanol step for protein precipitation on the particles, a lysis buffer was added and the mixture was

heated to 65 °C, inactivating the virus. The optimized non-automated protocol was implemented in the Hamilton Robotics Microlab STARlet liquid-handler aiming for full automation of sample preparation (Fig. 4). The digestion step, which bottlenecks sample preparation, was reduced to 2 h with no loss in sensitivity to the targeted peptides, resulting in a 4-h processing time for 96 samples.

**Qualitative analysis and protein absolute quantification.** Qualitative analysis of SARS-CoV-2 nucleoprotein was achieved by introducing [15]N-labeled chromogranin A as a surrogate standard. The peptide HSGFEDELSEVLENQSSQAELK was detected within the chromatographic window for SARS-CoV-2 and was

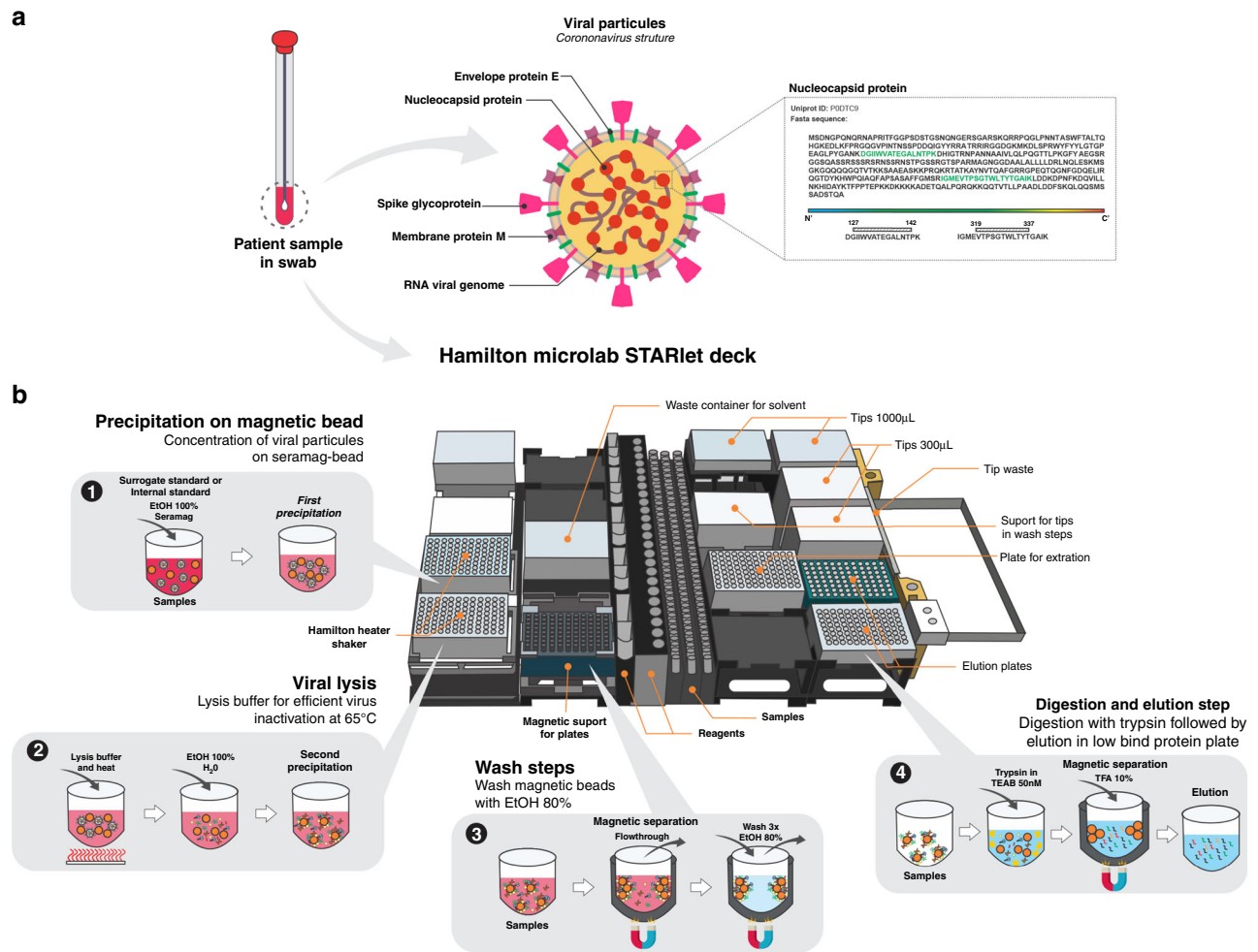

**Fig. 4 Overview of automated sample preparation for SARS-CoV-2 SP3-based extraction on a Hamilton Robotics Microlab STARlet liquid handling system. a** Patient samples were collected with swabs placed in virus transport medium or sterile saline solution and stored at −80 °C before automated sample processing. The highlighted nucleocapsid peptides were monitored by turbulent flow chromatography coupled to tandem mass spectrometry. **b** Schematic overview showing the steps and Hamilton desk setup for automated SP3 sample processing: (1) viral particle and internal standard precipitation on magnetic bead using ethanol (EtOH) to a final concentration of 50% (v/v); (2) one-step viral lysis using a sodium dodecyl sulfate-based (SDS) buffer and dithiothreitol (DTT) for 5 min at 65 °C followed by dilution and bead precipitation; (3) magnetic separation and three consecutive wash steps with ethanol 80% (v/v); (4) protein elution from magnetic bead followed by digestion with trypsin in triethylamonium bicarbonate buffer (TEAB) for 2 h at 37 °C, post-digestion acidification with trifluoroacetic acid (TFA), and recovery of peptides in a new sample plate.

used to check for protein digestion and to normalize IGMEVTPSGTWLTYTGAIK response in the Tier 3 assay (IGM/IS). Signal to background for IGMEVTPSGTWLTYTGAIK and DGIIWVATEGALNTPK were also calculated to classify specimens.

Recombinant unlabeled and [15]N-labeled SARS-CoV-2 nucleo-proteins were expressed in *E. coli*, purified and quantified by amino acids analysis. The incorporation yield of [15]N-labeled amino acids was found to be over 99% (see Supplementary Table 2). A calibration curve was produced by spiking the nucleoprotein into negative pooled samples in the range from 2 to 512 ng/mL. The concentration for the SIL standard was 10 ng/mL.

**Data analysis and analytical validation**. Analytical validation results are summarized in Table 3. The sensitivity and specificity of the Tier 1 and Tier 3 assays were determined against the real-time RT-PCR, considered the gold standard. For the Tier 3 assay, the receiver operating characteristics (ROC) curves were built to evaluate the performance of qualifiers for classifying samples. The limit of blank (LoB) for IGMEVTPSGTWLTYTGAIK and for

DGIIWVATEGALNTPK were used as cut-offs along with the cut-offs at the point of the maximum accuracy, as determined by ROC curves. Validation was performed with 80% (*n* = 432) of the dataset as training set and 20% as testing set (*n* = 108). Different predictive models were built using the qualifiers individually and a number of qualifiers combinations. The overall accuracies cal-culated with the training set showed that the combination of qualifiers S/N IGM (LoB cut-off = 1.65), S/N DGI (LoB cut-off = 0.85) and IGM/IS (ROC cut-off = 0.04) was the best predictive model. A testing set was used to confirm the performance of the predictive model. The ROC curve for the combination model provided an area under the ROC curve (AUC) of 0.91 (95% CI 0.84–0.98, Supplementary Fig. 5). Specificity was improved in the model compared to individual qualifiers, showing the importance of using two peptides to classify the samples (Supplementary Table 3). The combination of the three qualifiers, with specific cut-offs, allowed a distinction between positive and negative samples with an accuracy of 87.7% (sensitivity of 83.6% and specificity of 93.3%, Table 3 and Supplementary Table 3 and Table 3). The decision tree for this final predictive model is shown in Supplementary Fig. 6.

**Table 3 Summary of validation parameters for Tier 1 and Tier 3 assays. CI: confidence interval. CV: coefficient of variation. Source data are provided as a Source Data file.**

| Validation parameter | Results | |
|---|---|---|
| | Tier 1 | Tier 3 |
| Method comparison n Tier 1 = 445 samples n Tier 3 = 540 samples | Accuracy = 87.2% (CI 95%, 83.8–90.0%) Sensitivity = 78.2% (CI 95%, 72.4–83.0%) Specificity = 96.8% (CI 95%, 93.5–98.4%) | Accuracy = 87.7% (CI 95%, 79.4–95.2%) Sensitivity = 83.6% (CI 95%, 70.6–95.7%) Specificity = 93.3% (CI 95%, 82.4–100%) |
| Linearity n = 12, 4 levels | IGMEVTPSGTWLTYTGAIK: up to 1000 ng/mL $R^2$ = 0.9948, mean recovery = 101.2% DGIIWVATEGALNTPK: up to 1000 ng/mL $R^2$ = 0.9786, mean recovery = 103.6% | Not applicable |
| Limits of blank n Tier 1 = 52 n Tier 3 = 38 | IGMEVTPSGTWLTYTGAIK: relative response factor = 0.068 DGIIWVATEGALNTPK: relative response factor = 0.101 | IGMEVTPSGTWLTYTGAIK: Signal-to-noise = 1.65 DGIIWVATEGALNTPK: Signal-to-noise = 0.83 |
| Limits of detection n Tier 1 = 50 n Tier 3 = 20 | IGMEVTPSGTWLTYTGAIK: relative response factor = 0.117 (equivalent to 2.7 ng/mL) DGIIWVATEGALNTPK: relative response factor = 0.162 (equivalent to 3.2 ng/mL) | IGMEVTPSGTWLTYTGAIK: Signal-to-noise = 3.03 DGIIWVATEGALNTPK: Signal-to-noise = 1.10 |
| Limits of quantification n = 45 (5 levels, 9 independent replicates) | IGMEVTPSGTWLTYTGAIK = 6.0 ng/mL DGIIWVATEGALNTPK = 4.4 ng/mL | Not applicable |
| Reproducibility n Tier 1 = 44 n Tier 3 = 10 | Negative: undetectable Positive (mean = 21.9 ng/mL): IGMEVTPSGTWLTYTGAIK: CV = 17.5% DGIIWVATEGALNTPK: CV = 19.2% | Negative: undetectable Positive: IGMEVTPSGTWLTYTGAIK: CV = 19.8% DGIIWVATEGALNTPK: CV = 18.7% |
| Carryover n = 12 | IGMEVTPSGTWLTYTGAIK: 1.2% and 0.2% (first and second injection) DGIIWVATEGALNTPK: 4.0% and 0.7% (first and second injection) | |
| Stability n = 299 (5 replicates per condition), 95% CI | Sample storage stability: Saline: stable for all conditions (30 days, 21 °C, 4 °C, −20 °C and −80 °C) Virus transport medium: $p < 0.05$ at 21 °C to low viral load for all storage time Thermal test (90 °C for 5 min): Stable for all conditions | |
| Interference | Undetected (Supplementary Table 5) | |

For the Tier 1, samples were discriminated using the limit of detection (LoD) calculated by means of relative response factors, i.e., peptide peak area over SIL peak area ratios (Supplementary Fig. 6). The sensitivity and specificity of IGMEVTPSGTWLTYT-GAIK and DGIIWVATEGALNTPK are shown in Table 3 and Supplementary Table 4. Performance analysis demonstrates that accuracies do not vary substantially between them, but sensitivity and specificity showed to be slightly different (Table 3 and Supplementary Table 4). Combining the two peptides improved the specificity (96.8%) if compared to detection based solely on one peptide, without compromising the sensitivity. Considering the LoD of the two peptides, the Tier 1 method was able to distinguish positives and negatives samples with an accuracy of 87.2%, a sensitivity of 78.2%, and a specificity of 96.8%.

Repeatability measurements evaluated over ten days revealed coefficients of variation (CVs) lower than 20% for a positive sample in both assays (Supplementary Fig. 7 and Table 3). Retention times and peak areas reproducibility across five days for surrogate standard (Tier 3), SIL internal standards (Tier 1) and the endogenous beta-actin peptides are shown in Supplementary Figs. 8–10. The retention time for all monitored peptides is stable throughout the four channels and 96-injection batch. The peak areas for the SIL internal standard peptides presented

lower variability compared to the surrogate standard peptide (Supplementary Figs. 9, 10). Since a degree of variability of the surrogate standard peptide was observed, the batch effect was evaluated for Tier 3. As shown in Supplementary Fig. 11, a batch effect was not observed in this study. Tier 1 included an additional layer of quality control based on the monitoring of SYELPDGQVITIGNER, a beta-actin derived peptide used as an endogenous control for sample collection. As shown in Supplementary Fig. 9, the high variation of the peak areas reinforced the importance to establish cut-off points in order to accept, or reject, a result due to problems associated with sample collection.

Linearity was estimated for the Tier 1 in viral transport medium and in negative pooled samples at concentrations ranging from 0.1 to 1000 ng/mL. As shown in Supplementary Figs. 12, 13, peptides presented linearity up to 1000 ng/mL with five orders of magnitude in viral transport medium (Supplementary Fig. 12) and four orders of magnitude using negative pooled samples (Supplementary Fig. 13).

In order to determine the influence of temperature on the stability of targeted proteins, samples stored at 21 °C, 4 °C, −20 °C, and −80 °C were compared to samples kept in liquid nitrogen. Sterile saline samples were stable even at room temperature for up

30 days, whereas virus transport medium samples required storage at 4 °C (Table 3). The protein biomarker presented stability after treating samples at 90 °C for 5 min (Table 3).

To evaluate a correlation between the targeted proteomics approach with real-time RT-PCR cycle thresholds, Tier 3 qualifiers (S/N DGI, S/N IGM, and IGM/IS) and a number of molecules (i.e., obtained from samples with concentration equal or higher than LoQ) in Tier 1, were plotted against viral loads. Moderate Pearson's coefficients ($r = 0.6$–$0.76$) were observed for Tier 3 (Supplementary Fig. 14). For Tier 1, higher correlations were obtained ($r = 0.78$ and $r = 0.85$, DGIIWVATEGALNTPK and IGMEVTPSGTWLTYTGAIK, respectively) (Supplementary Fig. 15).

Interference from other viruses was analyzed by testing specimens previously characterized by molecular testing. Targeted peptides from SARS-CoV-2 were absent in monoinfection cases (coronavirus 229E, coronavirus NL63, parainfluenza 1, parainfluenza 4, influenza A/H1-2009, HMPV, and respiratory sincycial virus) as well as coinfection cases (coronavirus HKU1/rhinovirus/enterovirus, rhinovirus/enterovirus, and rhinovirus/enterovirus/HMPV) (Supplementary Table 5).

## Discussion

In response to the urgent need to develop alternative diagnostic tests for the novel coronavirus (SARS-CoV-2) pandemic, we developed a method that satisfies the following requirements: specificity towards SARS-CoV-2; good sensitivity compared to the gold standard real-time RT-PCR; sample preparation of high-volume processing in the shortest time possible; and fast acquisition by multiplexing liquid chromatography coupled to tandem MS. The first step of this process was to establish surrogate peptide targets for SARS-CoV-2 proteins. Specimens with high viral loads previously characterized by real-time RT-PCR enabled building a spectral library from data-dependent analysis that served as the basis for determining the most suitable peptide candidates. An initial 60-min PRM method explored 17 peptides from four proteins and revealed that nine nucleoprotein peptides were the most intense and therefore promising options for SARS-CoV-2 detection. Also, their sequences were unique among those deposited in UniProt and a local search for mutations showed that these regions are conserved.

Using microflow liquid chromatography coupled to hybrid quadrupole-orbitrap MS analysis we were able to complete sample acquisition in only 9 min. However, to achieve high-volume testing, we investigated alternatives to LC-MS/MS. Four-channel TFC coupled to triple quadrupole MS detection was able to meet this requirement by increasing the speed of analysis and by incorporating an additional on-line solid-phase cleanup with the dual-column approach. TFC-MS/MS has been used for high-throughput determination of several important clinical bio-markers, such as small metabolites and proteins[29,30]. The TurboFlow Cyclone-P HPLC $0.5 \times 50$ mm column used for TFC was selected for its robustness while requiring lower flow rates to generate turbulence[9]. From the initial panel of eight selected peptides, the two most hydrophobic peptides, DGIIWVATE-GALNTPK and IGMEVTPSGTWLTYTGAIK, yielded the best results in TFC coupled to triple quadrupole MS detection. Two steps limit the speed of dual-column TFC assays, namely transfer and gradient elution[31]. During the transfer step, the extraction solvent previously stored in the loop is delivered to the extraction column combined with a weak mobile phase from the elution pump to focus the analytes onto the analytical column. Under heated-electrospray ionization, DGIIWVATEGALNTPK and IGMEVTPSGTWLTYTGAIK provided best signal intensities and therefore were the ones selected, eluting within a 2.5 min window

and enabling multiplex by four samples within 10 min. Thus, the system can process more than 500 samples in 24 h.

System suitability tests[32] were monitored for each batch in addition to quality control checks. While control checks are used to evaluate the entire workflow, the system suitability process monitors retention time, peak resolution and width, signal stability, and instrument response, checking the performance of the LC-MS instrument before the analytical run. The use of Skyline, Panorama, and AutoQC software automated system suitability and simplified longitudinal data analysis[33]. These daily checks are critical to ensure data quality and to avoid batch effects that could mainly affect the qualitative strategy. As shown previously, batch effect was not observed in the Tier 3 assay.

Another layer of quality control was incorporated in the Tier 1 assay, which was the monitoring of an endogenous peptide derived from trypsin digestion of human beta actin. This control aims to verify biospecimen recovery and to be an additional checking point for protein extraction and digestion. Beta actin is also commonly used in real-time RT-PCR methods as a positive control for sampling, nucleic acid extraction, and amplification reaction[34]. The variation observed in the endogenous peptide areas among samples was expected due to the intrinsic variability in swab sample collection, which may yield varied amounts of the sample depending on the collecting technique, the individual's anatomy, virus shedding and clogging of the nostrils[35]. A cut-off point was established to minimize the probability of false-negative detection. On the other hand, the targeted proteomics whole workflow presented low variability, as can be inferred by monitoring the SIL standard peak areas across the analytical batches.

Carryover is a ubiquitous phenomenon in liquid chromatography whereby the analytes from a previous injection are retained by adsorption in the flow path within the LC-MS system and detected in the subsequent injections. In the specific case of qualitative detection of SARS-CoV-2 by targeted proteomics, carryover may represent one of the greatest analytical challenges because viral load can range over eight orders of magnitude and erroneous biomarker identification can lead to false-positive detections. Thus, a significant part of the efforts to develop this method was dedicated to minimizing carryover. After isolating the main sources of carryover within the LC-MS system, the TFC column was found to be responsible for the largest part of the carryover observed. Among cleaning solutions evaluated to reduce carryover in the TFC column, DMSO and TFE were the ones most effective. DMSO was previously indicated to reduce peptide adsorption when added to the sample solution and autosampler washing solutions[36]. Here, DMSO was included in the first autosampler washing solution and in the TFC mobile phase D. TFE was also previously reported to reduce peptide carryover in nano chromatography[37]. Here, a combination step alternating 20% DMSO and 2% TFE in water with an organic cleaning solution (acetonitrile/isopropanol/acetone, 40:40:20, v/v) was used to reduce carryover, resulting in a 4.5-min TFC column cleanup, which starts after analyte transfer to the chromatographic column. Nevertheless, as a carryover-free system is virtually unachievable, a rule was established through data processing where the succeeding two samples following a high-intensity sample should be reinjected.

To enable the analysis of hundreds of samples with a short turnaround time, we implemented a modified automated SP3 protocol. SP3 is based on the binding of proteins to paramagnetic beads in the presence of an organic solvent, followed by extensive washings and digestion[38]. Recently, Müller and colleagues described an automated SP3 implemented on an Agilent Bravo platform, which was applied to cell culture and formalin-fixed paraffin-embedded tissue sections[39]. Their method required 3.5 h to process 96 samples until reaching the digestion step, which

required an additional 4 h. With our strategy, all steps can be completed in the robotic platform and the resulting microplate is ready to be processed by TFC-MS/MS. However, one offline centrifugation step was included to accelerate the magnetic bead collection. The use of a robotic liquid handler not only reduced processing time but also decreased the risk of infection for the laboratory personnel.

Respiratory tract samples are intrinsically heterogeneous compared to biological matrices such as plasma and urine. Several factors influence virus load in respiratory tract samples such as method of collection, anatomical collection site (e.g., nasopharyngeal and/or oropharyngeal), type of swab, sampling at a given diagnostic window, and individual variability[40,41]. Here, we focused on a dependable approach to successfully detect the virus in respiratory tract specimens rather than determine the amount of proteins in these clinical specimens. Even though data on specificity and sensitivity of immunoassays targeted to virus proteins are still scarce, targeted proteomics most likely present higher specificity. Moreover, selectivity is an intrinsic feature for MS-based tests and combined with unique peptide sequences increases the potential application for this strategy[42,43].

The specificity of the Tier 3 was enhanced by combining three qualifiers: S/N IGM, S/N DGI and IGM/IS. Although our method is less sensitive than real-time RT-PCR, it detects 83.6% of positive cases with high specificity (93.3%). The intrinsic real-time RT-PCR sensitivity is currently not possible to be surpassed by any protein analysis, mainly because proteins cannot be multiplied. However, in a scenario where large-scale population testing is needed and the supply for real-time RT-PCR reagents and instruments is insufficient to cover this demand[4,5], targeted proteomics provides an alternative to complement real-time RT-PCR testing. Currently, the demand for routine tests based on LC-MS/MS dropped in clinical laboratories due to the several levels of social distancing necessary to limit the spread of SARS-CoV-2 around the world. Therefore, there are thousands of idle LC-MS/MS instruments that could be used for SARS-CoV-2 testing.

The Tier 1 assay based on PSAQ (protein standard absolute quantification) strategy[44] incorporated recombinant NCAP, which was quantified by amino acid analysis[45] and also by a SIL NCAP, allowing protein absolute quantification[44]. The use of SIL standards controls and corrects inter-sample recovery variance such as those arising from differences in protein capture by the magnetic particle, digestion efficiency, and matrix effect, leading to a higher confidence detection and assay precision[27]. The determination of peak boundaries is more objective in Tier 1 than Tier 3 since it is guided by their respective intra sample heavy peptides. In Tier 3, it is important to verify external references such as quality control and system suitability samples. Therefore, the time required for data processing is reduced in Tier 1. Limits of quantification evaluated in the Tier 1 assay for peptides IGMEVTPSGTWLTYTGAIK and DGIIWVATEGALNTPK were 6.0 and 4.4 ng/mL, respectively (Table 3). Huillet and colleagues[16] achieved a limit of quantification of 5.5 ng/mL for troponin I using a PSAQ approach based on analyte immunoenrichment and fractionation by electrophoresis. Therefore, a combination of the modified SP3 protocol with TFC not only provided good sensitivity for biomarker detection in complex matrices but also enabled the analysis of hundreds of samples in a short turn-around time. Despite the advantages of Tier 1 previously mentioned, we found no significant difference between Tier 1 and 3 for the most relevant parameter, which is the diagnostic accuracy, since we observed an overlapping in the confidence intervals for sensitivity and specificity in both Tiers.

The automated sample preparation reduces the variability associated with manual sample processing. The reproducibility evaluated for the entire workflow for both Tier 1 and 3 presented CVs inferior to 20% and in accordance with good practices proposed to clinical proteomics assays[46]. Linearity was verified in five orders of magnitude when the NCAP protein was diluted in viral transport medium (Supplementary Fig.12). As expected, due to matrix effect, the linearity was reduced to the range from 1 to 1000 ng/mL when negative pooled samples were used as diluent (Supplementary Fig. 13). Some strategy to reduce matrix complexity such extra steps for analyte enrichment would be necessary to reach even lower limits of detection. In the dataset analyzed, <4% of the positive samples were above the upper range of linearity.

An additional positive aspect of targeted proteomics over real-time RT-PCR is the analyte stability. While it is recommended to store specimens for RNA testing at −80 °C right after sample collection[47], the targeted protein is stable in saline solution at room temperature for up to 30 days and therefore samples can be collected even in situations where their storing at −70 °C is not an option, like in remote areas. The targeted peptides can be detected after thermal inactivation at 90 °C for 5 min. It has been demonstrated that disinfection at 80 °C for 1 min was effective to reduce coronavirus infectivity[48]. Thus, adding the thermal inactivation step would not affect the results and it would be an extra safe step to reduce the biologic risk for infection of laboratory staff during testing. Lastly, our targeted proteomics SARS-CoV-2 testing is approximately half the cost of real-time RT-PCR.

It is important to note that similarly to what would happen with a test based on real-time RT-PCR, targeted proteomics is susceptible to false-negative detection in case a mutation occurs in the coding regions of the selected peptides. Any amino acid sequence modification, except for leucine/isoleucine replacements, would modify the peptide molecular mass turning it undetectable by predefined mass to charge coordinates. In addition, mutation in sequences preceding or succeeding the targeted peptide coding region could modify trypsin cleavage sites. Targeted proteomics has been successfully used to identify and quantify mutant proteins[49], but the targeted mass transition must be redesigned. A study comparing thousands of SARS-CoV-2 genomes sequences indicated Orf1a, Nsp11, Nsp13, and the Spike protein as the genomic regions with the strongest signal of recurrent mutation sites[28]. Although not detected locally, this study indicated potential mutations on DGIIWVATEGALNTPK and IGMEVTPSGTWLTYTGAIK coding regions. As both peptides presented similar intensities, the absence of one of them could indicate the presence of a mutation and the sample should be confirmed by real-time RT-PCR.

Our study has another limitation that should be acknowledged. The MS method relies on different reagents and instruments that may or may not be available during the pandemic. Some materials, such as swabs, are also used for real-time RT-PCR methods and could be limited due to high demand. Furthermore, large-scale application of targeted proteomics would require standardization across laboratories by reference material and the availability of SIL standards. The material produced by this study such as calibration curves and SIL protein could be a candidate for the method standardization.

In conclusion, here we present a proof of concept application of automated sample preparation and multiplexing TFC coupled to triple quadrupole MS as a feasible alternative for detecting SARS-CoV-2 in clinical respiratory tract samples in a large scale at a population level. Our strategy enables high-volume testing in a short turnaround time and can be combined with other tests currently used for the detection of COVID-19 infection to help control the pandemic.

## Methods

**Clinical samples.** Respiratory tract samples (combined materials from nasopharyngeal and oropharyngeal swabs) were collected in virus transport medium[50] or sterile saline solution and stored at −80 °C. All specimens used in this study were previously analyzed by an in-house real-time RT-PCR method implemented according to WHO guidelines[51]. A total of 985 respiratory tract samples (540 real-time RT-PCR positive and 445 real-time RT-PCR negative) were included in the study. Ten positive samples with low-cycle threshold values were used to create five pools for shotgun bottom-up proteomics analysis. Cross-reactivity was evaluated against specimens of other human coronaviruses (HCoV-HKU1, HCoV-229E, and HCoV-NL63), Influenza A (H1N1), respiratory syncytial virus (RSV), HMPV, parainfluenza virus types 1 and 4, and rhinovirus/coronavirus HKU1/enterovirus coinfection previously characterized by Biofire FilmArray Respiratory Panel (bio-Mérieux, Marcy-l'Étoile, France). This study was approved by the Instituto Fleury Ethics Committee (Plataforma Brasil Certificate of Presentation for Ethical Consideration: 31686420.6.0000.5474). Patient identification was not recorded or registered. Informed consent from all patients/donors was waived because only specimens deidentified and collected as part of standard diagnostic protocols that would normally be discarded were included in this study.

**Production of SARS-CoV-2 recombinant nucleoprotein.** The nucleoprotein (2019-nCoV-N, NCAP_WCPV) gene, codon-optimized for expression in *E. coli*, was synthesized and subcloned under T7 promoter control into pET28a(+) vector at SacI and NotI restriction sites (GenScript, Hong Kong). The resulting plasmid 2019-nCoV-N_pET-28a(+) was used to transform *E. coli* BL21-AI competent cells.

Unlabeled protein was produced by growing the cells in LB broth with 30 μg/mL kanamycin at 37 °C, 250 rpm to an optical density at 600 nm of 0.6. Gene expression was induced with 0.2% arabinose for 4 h. Expression of $^{15}$N-labeled nucleoprotein was performed by inoculating 100 μL of a bacterial culture grown in LB into 500 mL of M9 minimal medium containing 1% $^{15}$NH$_4$Cl at 37 °C and 250 rpm. After arabinose induction, cells were grown at 25 °C, 250 rpm for 16 h. Cells were harvested by centrifugation at $3200 \times g$ for 10 min and pellets were resuspended in a buffer containing 50 mM Tris-HCl (pH 8.5), 300 mM NaCl, 10% glycerol, 0.1 mM PMSF and 0.4 mg/mL lysozyme. After a 20 min incubation, cell suspensions were disrupted by sonicating using four cycles of 15 s at 80% energy (UP100H, Hielscher, Teltow, Germany) on an ice bath and the lysates were then centrifuged at $20,000 \times g$, at 4 °C for 15 min.

Supernatants were purified on an ACQUITY UPLC H-Class Bio System (Waters, Milford, MA, USA) with HiTrap TALON Crude column (Cytiva, Uppsala, Sweden) equilibrated with binding buffer (20 mM sodium phosphate pH 8 and 0.5 M NaCl), washed with binding buffer at 0.5 mL/min for 10 min and then eluted with a linear gradient from 0 to 0.225 M imidazole in binding buffer in 10 min. Recombinant protein-containing fractions were pooled and dialyzed overnight in phosphate-buffered saline pH 7.3 at 4 °C. Protein purity was determined by SDS-PAGE and protein quantification was achieved by amino acid analysis after hydrolysis with 6 M hydrochloric acid for 24 h at 110 °C. $^{15}$N-Labeled incorporation efficiency was verified by digesting 10 ng/mL $^{15}$N-labeled nucleoprotein and analyzing by the targeted method as described below. The purified protein was supplemented with 30% glycerol and stored at −80 °C until use.

**Nonautomated sample processing.** Two hundred microliters of clinical specimens were transferred to 1.5-mL conical polypropylene tubes. Proteins were precipitated by the addition of 5 volumes of ethanol, followed by storage at −80 °C for 30 min and centrifugation at 4 °C, $20,000 \times g$ for 15 min. Proteins were digested with trypsin using a modified single-pot solid-phase-enhanced sample preparation (SP3) protocol described by Hughes[38]. In brief, after careful removal of supernatant by aspiration, the pellets were suspended in 50 μL of lysis buffer (1% SDS, 5 mM DTT in 50 mM TEAB pH 8.5) and lysed and reduced at 85 °C, 2000 rpm for 20 min in a thermomixer. Next, 20 μL of Sera-Mag magnetic carboxylate modified particles (GE Healthcare, Little Chalfond, UK), prepared by combining equal volumes of hydrophilic and hydrophobic particles washed with an equal volume of water and resuspended in water to reach a final concentration of 2 mg/mL, was added to the tubes followed by one volume of ethanol, and the mixture was incubated in a thermomixer at room temperature, 1000 rpm for 10 min. The beads were immobilized on a magnetic rack, the supernatant was removed, and the beads were washed three times with 200 μL of 80% ethanol. A total of 75 μL of trypsin (Gold Mass Spectrometry Grade; Promega, Madison, WI, USA) at 66.7 μg/mL in 50 mM TEAB pH 8.5 was added and the mixture was incubated overnight at 37 °C, 500 rpm in a thermomixer. Lastly, 5 μL of 10% TFA in water was added, the beads were immobilized on a magnetic rack, and the tryptic digest transferred to total recovery glass vials.

**Automated sample processing.** Automated sample preparation was achieved on a Hamilton Robotics Microlab STARlet liquid handling system (Hamilton Company, Reno, NV, USA) equipped with eight pipetting channels, 96-channel multiprobe head, labware gripper, and an automated heater shaker. The robotic liquid handler was modified with a HEPA (high-efficiency particulate arrestance) filter connected to an exhaustion pump. Programming and operation were achieved using Hamilton Robotics Venus Three software. Samples to both protocols (Tier 1 and Tier 3) were extracted equally except for the distinct internal standards. Two hundred microliters of clinical specimens were transferred to a 96-deep-well plate. Next, 30 μL of fully $^{15}$N-labeled nucleoprotein internal standard at 80 ng/mL (Tier 1 assay) or $^{15}$N-labeled chromogranin A surrogate standard at 15 μg/mL (Tier 3 assay) was dispensed into the plate followed by 50 μL of Sera-Mag carboxyl modified magnetic particles suspension in water at 1 mg/mL. One volume of ethanol was then dispensed into the plate and the mixture was agitated at 900 rpm for 5 min. Samples were lysed and reduced with 50 μL of lysis buffer and incubated at 65 °C at 1000 rpm for 5 min. After lysis, an additional 100 μL of water and 150 μL of ethanol were added and the plate was agitated at 900 rpm for 5 min. The plate was spun down to accelerate the bead separation process and transferred to a Magnum EX Universal Magnet Plate (Alpaqua, Beverly, MA, USA) for 5 min where the supernatant was removed. The immobilized beads were washed once with 800 μL of 80% ethanol and twice with 200 μL of 80% ethanol. A solution of TPCK-treated trypsin (75 μL at 65 μg/mL; Sigma-Aldrich, St. Louis, MO, USA) in 50 mM TEAB pH 8.5 was dispensed and the plate was incubated at 37 °C, 1000 rpm for 2 h. Lastly, 5 μL of 10% TFA in water was added to each well and after short mixing, the beads were immobilized on a magnetic rack. Tryptic digests were transferred to a Protein LoBind Deepwell plate 96/500 μL. The target plate was removed from the robotic liquid handler and stored at −20 °C until analysis by liquid chromatography-tandem mass spectrometry (LC-MS/MS).

**Untargeted LC-MS/MS analysis.** Target peptide selection was achieved on an UltiMate 3000 Nano LC system coupled to a Q-Exactive HF-X mass spectrometer (Thermo Fisher Scientific, Bremen, Germany) via an EASY-Spray source operating in positive ion mode. The UltiMate 3000 Nano system was fitted with a Pep-Map100 C$_{18}$ 5 μm, 0.3 × 5 mm sample trapping pre-column and a PepMap RSLC C18 2 μm, 150 μm × 15 cm analytical column (Thermo Fisher Scientific).

DDAs were obtained injecting 10 μL of the tryptic digest loaded in the trapping column with 0.1% TFA at 150 μL/min for 3 min. For chromatographic separation, the flow rate was 1.5 μL/min and the column was maintained at 45 °C; solvent A was 1% DMSO, 0.1% formic acid in LC/MS grade water and B was 1% DMSO, 0.1% formic acid in acetonitrile. A 60-min linear gradient was used as follows: 3–20% B for 50 min, 20–40% B for 10 min, 40–90% B for 2 min. Source parameters were set as follows: spray voltage = 2.2 kV, capillary temperature = 275 °C, and S-lens RF level = 50. The MS spectra were acquired with Xcalibur software (version 4.2.47, Thermo Fisher Scientific) with 120,000 mass resolution ($m/z$ 200) from $m/z$ 350 to 1650, AGC target of $3 \times 10^6$, and maximum injection time (IT) of 60 ms. The MS/MS spectra were acquired for the 15 most intense ions of each MS scan (TopN = 15) with 15,000 mass resolution ($m/z$ = 200), an isolation window of $m/z$ 1.6, automatic gain control (AGC) target of $2 \times 10^5$, maximum IT of 60 ms, and (N) CE = 27. Single-charged ions and those with more than six charges were excluded and a 20-s dynamic exclusion was used. The signal at $m/z$ 401.92272 from DMSO was used as a lock mass.

**Selection of target peptides.** DDA raw files were processed by the MaxQuant software version 1.6.14[52] and searched against the UniProt SARS-CoV-2 pre-release (downloaded on 13 March 2020). Mass tolerance values for MS and MS/MS and the false discovery rate were set at 20 ppm and 1%, respectively. Methionine oxidation and N-terminal acetylation as variable modifications.

Skyline (Daily version 20.1.9.234)[53] was used to build a spectral library from data processed by MaxQuant. The first set of candidate peptides was established importing UniProt SARS-CoV-2 pre-release into Skyline. Only peptides matching the library, fully digested and with no cysteine residues were included. Filtered peptides were exported into an isolation list to construct a parallel reaction monitoring (PRM) method for the mass spectrometer (Supplementary Fig. 1). Chromatographic and ion source parameters were identical to those described above. PRM data were acquired with 120,000 mass resolution ($m/z$ 200), AGC target of $3 \times 10^4$, maximum IT of 250 ms, isolation window of $m/z$ 1.6, and (N)CE = 27. Positive and negative samples were analyzed by the PRM method loaded into Skyline and the number of targets was reduced to the top 17 most intense ones across positive samples and absent in negative samples.

A homology search for targeted peptides was performed using blastp against SwissProt Uniprot[54] databases (retrieved on 28 March 2020).

SARS-CoV-2 sequences were downloaded from the GISAID (gisaid.org) platform on 14 April 2020. Searches were filtered from location (South America/Brazil) and only those sequences with full coverage on CDS coding region for nucleocapsid were included. The sequences were aligned together with NCBI Reference Sequence NC_045512.2 by the Clustal Omega server[55] and visualized in JalView alignment editor (version 2.11.0)[56]. The frequency of single-nucleotide polymorphisms in the nucleoprotein coding region was verified by genome alignment of the 7666 SARS-CoV-2 genomes using SARS-CoV-2 Alignment Screen[28]. Amino acid variation including substitutions, insertions and deletions, was retrieved from GISAID hCoV-19 sequences database (last update 11 Aug 2020) through CoV-GLUE (cov-glue.cvr.gla.ac.uk) for DGIIWVATEGALNTPK and IGMEVTPSGTWLTYTGAIK. Three nucleotides located before and after the corresponding coding regions were included to check for modifications in trypsin cleavage sites.

**Fast separation PRM method.** Fast PRM acquisitions were achieved with the following chromatographic separation process: samples were loaded into the trapping column with 0.1% TFA in water at 150 μL/min for 30 s; the flow rate was 1.5 μL/min and the column was maintained at 45 °C; solvent A was 1% DMSO, 0.1% formic acid in LC/MS grade water and B was 1% DMSO, 0.1% formic acid in acetonitrile. A 7-min linear gradient was used as follows: 24–25% B for 5 min, 25–80% B for 12 s keeping at 80% B for 30 s and returning to 24%. PRM data were acquired with 120,000 mass resolution ($m/z$ 200), AGC target of $3 \times 10^4$, maximum IT of 250 ms, isolation window of $m/z$ 1.6, and (N)CE = 27.

**TFC coupled to MS detection.** A Transcend TLX-4 system consisting of four Dionex UltiMate 3000 quaternary pumps, four Dionex UltiMate 3000 binary pumps, one VIM, and one CTC PAL autosampler was coupled to a TSQ Altis Triple Quadrupole Mass Spectrometer fitted with a heated-electrospray ionization (HESI) source (Thermo Fisher Scientific, San Jose, CA, USA). Aria MX (version 2.5, Thermo Fisher Scientific) was used to control the system and acquisition was done with TraceFinder software (version 4.1, Thermo Fisher Scientific). The TLX-4 system was fitted with four TurboFlow Cyclone-P HPLC 0.5 × 50 mm columns (Thermo Fisher Scientific) and four Acquity UPLC BEH C₁₈, 1.7 μm, 2.1 mm × 50 mm columns (Waters, Milford, MA, USA). The mobile phase for the first dimension was 0.5% acid formic in water (mobile phase A), acetonitrile (mobile phase B), acetonitrile/isopropanol/acetone (40:40:20, v/v) (mobile phase C), and 20% DMSO/2% TFE in water (mobile phase D). The mobile phase for the second dimension was 0.1% acid formic, 1% DMSO in water (mobile phase A) and 0.1% acid formic, 1% DMSO in acetonitrile (mobile phase B).

High-throughput screening acquisitions were obtained by injecting 25 μL of the tryptic digest sample onto the TurboFlow column with 0.5% acid formic in water at 1.2 mL/min. The flow was then reversed and slowed, and the retained peptides were eluted and transferred onto the analytical column. The total run time was 10 min, but multiplexing enabled a fourfold reduction in the overall analysis time. TSQ Altis optimized parameters were set as follows: spray voltage (kV): +4.0, sheath gas pressure (arb): 60, auxiliary gas pressure (arb): 15, sweep gas pressure (arb): 2, ion transfer tube temperature (°C): 300, vaporizer temperature (°C): 200, Q1 Resolution (FWHM): 2.0, Q3 Resolution (FWHM): 2.0, and CID gas (mTorr): 1.5. SARS-CoV-2 selected peptides (DGIIWVATEGALNTPK and IGMEVTPSGTWLTYTGAIK), ¹⁵N-labeled chromogranin A surrogate standard (heavy HSGFEDELSEVLENQSSQAE LK), ¹⁵N-labeled nucleoprotein (heavy DGIIWVATEGALNTPK and IGMEVTPSGTWLTYTGAIK) and human beta actin (SYELPDGQVITIGNER) peptides were detected using SRM. Three transitions were monitored for each peptide (Table 4). Collision energy (CE) and RF lens voltage (RF) for all peptides were optimized using the Skyline optimization pipeline[57].

**Data processing and interpretation.** Data processing was done using Skyline[58] and system suitability using Panorama (version 18.2) and AutoQC Loader (version

1.1.0.18345)[33]. In brief, the raw data were imported, peak integration was reviewed individually, and the results were exported into a spreadsheet along with the surrogate standard peak areas, internal standard peak areas, targeted peptide peak areas, and background noise. The raw data were processed without any transformation (i.e., smoothing). For Tier 3, the analyte to surrogate standard peak area ratio (IGM/IS and DGI/IS) and signal-to-background or signal-to-noise ratio (S/N IGM and S/N DGI) were calculated for each targeted peptide. The calculated qualifiers (S/N IGM, S/N DGI, IGM/IS and DGI/IS) were used to build the predictive models, which were obtained comparing accuracies from manual combinations assisted by the qualitative method comparison module from EP Evaluator (version 12). For Tier 1, calibration curves in virus transport medium were prepared using unlabeled protein purified as previously described here. The analyte peak areas to the SIL internal standard peak areas ratio were calculated for each targeted peptide and then converted to concentrations using calibration curves with linear in log space regression. Positive and negative samples were discriminated according to the sensibility achieved for the assay using the limits of detection calculated by means of relative response factors. The system suitability performance and quality control analysis were evaluated through Skyline software[58,59]. Positive and negative quality controls were included in each run. If the control material failed to yield the expected results, the run was rejected. The entire workflow for data processing and interpretation for Tiers 1 and 3 is summarized in the Supplementary Fig. 6.

**Method validation.** Method analytical validation was based on the Clinical Laboratory and Standards Institute guideline[60] and in the best practice acceptance criterion for quantitative LC-MS/MS-based protein assays[27,46]. Sensitivity and specificity were established by comparison with the in-house real-time RT-PCR method for SARS-CoV-2. The total number of specimens analyzed for comparative studies was 540 for Tier 3 (311 positives and 229 negatives by real-time RT-PCR) and 445 (229 positives and 216 negatives by real-time RT-PCR) for Tier 1. Interference was assessed using clinical specimens from other human coronaviruses (HCoV-HKU1, HCoV-229E, and HCoV-NL63), influenza A (H1N1), rhinovirus, enterovirus, RSV, HMPV, and parainfluenza virus types 1 and 4. LoB was estimated by measuring replicates of a negative sample (no signal detected by real-time RT-PCR) and limit of detection (LoD) by measuring pools of samples with low viral load (as determined by real-time RT-PCR) for Tier 3 and negative pooled samples spiked with low concentrations of the unlabeled protein for Tier 1[61,62]. Limit of quantification (LoQ) was determined for Tier 1 along 10 days using negative pooled samples spiked at five different concentrations of the unlabeled protein and was defined by the lowest concentration resulting in a CV <20%[61]. Linearity range was estimated for Tier 1 using viral transport medium spiked with unlabeled protein from 0.1 to 1000 ng/mL and negative pooled samples spiked with unlabeled protein from 1 to 1000 ng/mL[46]. Calibration curves were produced using the viral transport medium spiked with unlabeled protein from 0 to 512 ng/mL. Reproducibility was evaluated using negative and positive pools over ten days and

**Table 4 Selected reaction monitoring (SRM) parameters for determination of SARS-CoV-2 targeted peptides. DGI: DGIIWVATEGALNTPK; IGM: IGMEVTPSGTWLTYTGAIK; HSG: HSGFEDELSEVLENQSSQAELK; and SYE: SYELPDGQVITIGNER.**

| Assay | Peptide | Start time (min) | End time (min) | Precursor (m/z) | Product (m/z) | Collision energy (V) | RF Lens (V) |
|---|---|---|---|---|---|---|---|
| Tier 3 | DGI (+2) | 1.7 | 2.3 | 842.948 | 1001.526 | 25 | 160 |
| | | | | | 1100.595 | 26 | |
| | | | | | 1286.674 | 25 | |
| | IGM (+2) | 2.2 | 2.6 | 1013.021 | 1394.731 | 31 | 170 |
| | | | | | 1495.779 | 31 | |
| | | | | | 1594.847 | 31 | |
| | HSG (heavy) (+3) | 0.7 | 1.7 | 835.359 | 1017.462 | 27 | 170 |
| | | | | | 1143.423 | 27 | |
| | | | | | 1243.489 | 27 | |
| Tier 1 | DGI (+2) | 1.0 | 2.0 | 842.948 | 1001.526 | 25 | 160 |
| | | | | | 1100.595 | 25 | |
| | | | | | 1286.674 | 25 | |
| | DGI (heavy) (+2) | 1.0 | 2.0 | 852.420 | 1013.49 | 25 | 160 |
| | | | | | 1113.556 | 25 | |
| | | | | | 1301.629 | 25 | |
| | IGM (+2) | 1.5 | 2.5 | 1013.021 | 1394.731 | 31 | 170 |
| | | | | | 1495.779 | 31 | |
| | | | | | 1594.847 | 31 | |
| | IGM (heavy) (+2) | 1.5 | 2.5 | 1023.490 | 1409.687 | 31 | 170 |
| | | | | | 1511.731 | 31 | |
| | | | | | 1611.797 | 31 | |
| | SYE (+2) | 0 | 1.4 | 895.949 | 901.51 | 25 | 170 |
| | | | | | 1201.617 | 25 | |
| | | | | | 1298.67 | 25 | |

the statistical analysis was performed using MSstats plugin for Skyline[63]. The system carryover was analyzed by injections of high viral load samples followed by three sequential injections of blank samples; the peptide area of blank samples was then compared to the peptide area of high-intensity samples. The stability study used sterile saline and virus transport medium pools with low and high viral loads samples stored at 21 °C, 4 °C, −20 °C, −80 °C and in liquid nitrogen for up to 30 days. All conditions were analyzed in quintuplicate in the same run and t-tests were performed between each condition and controls stored in liquid nitrogen. Stability after thermal inactivation was evaluated by heating samples at 90 °C and then comparing with non-heated samples.

Statistical analysis was performed using Excel, EP Evaluator (version 12), and R (version 3.6.0; packages ggplot2 and pROC)[64–66]. The sample size and confidence intervals used to qualitative method comparisons were based on the Clinical and Laboratory Standards Institute (CLSI) guidelines[60]. The overall diagnostic accuracy of the classification system was determined by calculating the area under the ROC curve. Confidence intervals were based on bootstrap sampling algorithms. Principal component analysis (PCA) was applied to analyze potential batch effects with samples plotted according to their day of processing. Two-sided unpaired t-tests were applied to investigate the differences between groups in stability study with p value <0.05 as the threshold for defining significance. Correlations with real-time RT-PCR were conducted using the Python/Scikit-learn library[67]. Viral loads were estimated using the equation $y = 3 \times 10^{12} e^{-0.693x}$, where x is the cycle threshold value and y is the estimated viral load (number of virus copies).

**Material availability**. Calibration curve aliquots, vectors, and/or transformed bacterial clones for isotope standard production and method standardization may be provided under request.

**Reporting summary**. Further information on research design is available in the Nature Research Reporting Summary linked to this article.

## Data availability

The mass spectrometry untargeted proteomics data (raw files and spectral library) have been deposited to the ProteomeXchange Consortium via the PRIDE[68] partner repository with the dataset identifier PXD021328. Targeted proteomics data (PRM 60- and 9-min analyses, SRM analyses and summary datasheet) are available through the Panorama repository[59] with the dataset identifier PXD019300 and via https://panoramaweb.org/labkey/Fleury_SARS-Cov-2.url. SARS-CoV-2 protein sequence information used in this study is available from UniProt (https://covid-19.uniprot.org). Multiple sequence alignments of SARS-CoV-2 genomes (as of 14 April 2020 and 11 August 2020) are available from GISAID through CoV-GLUE (http://cov-glue.cvr.gla.ac.uk/). Reference genome sequence used in this study was obtained from the National Center for Biotechnology Information (NCBI) with GenBank accession NC_045512.2. All other data generated are included in Figures and Tables in this published article. Source data are provided with this paper.

## Code availability

R and Python scripts are available at https://github.com/FleuryRD/COVID19MS.

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

## Acknowledgements

We thank Daniel Wagner Fonteles Alves for data analysis, Lucia Sivieri de Assis Rocha, Lina Yoshida, and David Domingues Pavanelli for helpful discussions, Gabriela Rampazzo Valim, Luciana Guilhermino Pereira and Adauto Ayala for excellent technical assistance, Tiago Mendonça dos Santos for statistical analysis, and the editors at Publicase for reviewing this manuscript.

## Author contributions

V.M.C., K.H.M.C., L.G.V., A.L., R.A.S., and G.G.O. conceived and designed the experiments; K.H.M.C., L.G.V., A.L., R.A.S., G.G.O., and V.M.C. performed research; A.N.O. and A.M.F. contributed the clinical samples and analytical data; K.H.M.C., L.G.V., A.L., R.A.S., G.G.O., V.M.C., C.S.L., and C.F.H.G. analyzed data; R.A.S created Figs. 1, 2 and 4; V.M.C., K.H.M.C., L.G.V., A.L., R.A.S., and G.G.O. wrote the paper; A.M.F., C.F.H.G., and M.C.T.P. critically revised the manuscript. All authors have read and agreed with the final version of the manuscript.

## Competing interests

The authors declare no competing interests.
