## [Peer Review File · Nature Communications]

REVIEWER COMMENTS

Reviewer #1 (Remarks to the Author):

Review of manuscript NCOMMS-20-19160

Bottlenecks in the availability and other challenges in viral testing for SARS-Cov-2 have raised the idea that mass spectrometry might complement RT-PCR-based testing. To my knowledge, the authors of this manuscript present the first targeted MS assay for SARS-Cov-2 to be deployed on a large number of clinical specimens; and the cross-comparison with RT-PCR results is important. Although they do acknowledge the “qualitative” nature of the assay, there are significant limitations of the “Tier 3” approach (as defined by Carr et al. MCP 2014). Importantly, the assay lacks trypsin-cleavable stable isotope-labeled internal standards for the NCAP protein that are needed to achieve the highest confidence and assay precision and to control for matrix effects on chromatography and digestion. These and other criteria for a peptide-based clinical diagnostic have been defined in Carr et al. and in Grant and Hoofnagle (Clin Chem 2015), and elsewhere, and the authors should more clearly state the significant limitations of this work in the context of these best practices. Nonetheless, this manuscript represents an important step forward in the development of a MS-based assay for SARS-Cov-2, particularly in the demonstration that viral proteins can be detected in respiratory sample types that are currently used for PCR-based assays. While the demonstration of throughput is important, it should not outweigh concerns about assay sensitivity and specificity. The authors should address these specific comments:

1. The claim of assay cost does not belong in the title or abstract of this paper. There is no evidence presented that a MS assay can be performed at lower cost than RT-PCR, which would require a detailed accounting of capital equipment and operational (upkeep, personnel) costs, in addition to reagent costs. An emphasis in the title of the “proof-of-concept” nature of this work would be more appropriate.

2. The abstract mentions a shortage of reagents and instruments as a limitation to initial large-scale screening. These problems would likely have occurred for an MS test as well, and the lack of availability of sampling devices (e.g. swabs) and PPE for health care workers would be no different for these two technologies. There should be no debate that translation of an assay across laboratories would require the production of large amounts of reference material (e.g. recombinant NCAP protein) and stable isotope-labeled standards. Both of these would take considerable time and resources. These limitations will clearly need to be acknowledged in the manuscript.

3. The abstract should clearly state the qualitative nature of the assay and that it is a “Tier 3” assay based on the guidelines of Carr et al. (MCP 2014). The description of the “¹⁵N-labeled global standard” is somewhat misleading since it can be interpreted as a NCAP standard. Rather, this should be replaced with “surrogate standard” which more accurately describes this chromogranin a standard.

4. If word limits permit, it would be informative, in the abstract, to specify the respiratory fluid types analyzed (nasopharyngeal and oropharyngeal swabs, and nasal wash) and the n of each that was included in the validation study. Abbreviations, including UPLC and MS/MS should be avoided if possible.

5. The introduction should include peer-reviewed citations or for the shortage of SARS-Cov-2 tests (rather than “lay media” like the New Yorker), and the discussion of serological testing should have appropriate citations. The authors should acknowledge (and provide citations for) significant prior art with respect to clinical peptide-based assays that are either in use or under development, including those for thyroglobulin and apolipoproteins. In this context, the authors should also cite the guidelines for best practices referenced above.

6. The viral assay for HMPV used a 60 min assay (w/30 min gradient), not a 90 min assay as described in the introduction. The authors should also note that similar approaches have been extended to the identification of bacterial pathogens in respiratory tract samples (PMID: 31941798; another example of a “proof-of-concept” that did not include stable isotope-labeled internal standards for analytes of interest).

7. Throughout the paper, the authors should be specific about the type(s) and numbers of respiratory tract samples used at various steps. There are mentions of nasopharyngeal swab, oropharyngeal swab and nasal wash. The methods for collection and storage of each of the sample types should be clearly defined. For example, how many and what samples were stored in virus transport media (please cite the source) versus sterile saline.

8. The availability of both DDA and PRM/SRM data for re-analysis by the scientific community is noted. Could the authors also provide data for all of the validation studies and provide metadata, specifically the clinical sample type, for all of the individual samples?

9. A re-analysis of the DDA data in gpmdb (<https://gpmdb.thegpm.org>) confirms the identification of SARS-Cov-2 NCAP but also shows that cysteine alkylation was ineffective. This could be due to a lack of excess IAM versus DTT (5 mM of each was used). This should be noted although it does not affect

that overall results and subsequent steps used a reduction step (w/o alkylation). In addition, the .blib file in ProteomeXchange should not be the “redundant” version as this will not import into Skyline.

10. The potential for mutations in SARS-Cov-2 that compromise the long-term utility of the assay are a concern. In addition to the analysis presented by the authors as supplemental data, there have now been a number of reports that have documented known mutations over much larger datasets that should be referenced (e.g PMID: 32092483 and 32387564). It is worth noting that the P344S mutation in NCAP is immediately upstream of the cleavage site that generates the IGMEV peptide. Can the authors comment on whether this might affect the cleavage by trypsin? Based on your experience, can you comment in the discussion whether mass spectrometry assays would be better or worse than RT-PCR in the case of a rapidly mutating virus?

11. The SRM data should be included as a single Skyline file rather than split into a different file for each batch. In addition, while we understand the rationale for using a hybrid (small molecule and peptide) mode in Skyline, all of the data should be imported in peptide mode with a representative spectral library. This would enable calculation of a dotp for each peptide. If normalization to a global standard (chromogranin a) cannot be performed in peptide mode, this could be performed outside of Skyline.

12. The lack of stable isotope-labeled internal standards is a major limitation of the approach that needs to be discussed in detail. As mentioned above, clinical MS assays are expected to utilize spike-in of SIL peptide (winged, cleavable) or SIL protein prior to trypsin digestion. Despite the validation data presented, there is the potential for false-positive detection based on isobaric interferences. Without the internal standard, there is the potential for a false negative when a peptide has gone undetected because it has fallen outside of the measured retention time (this could be a rationale for excluding the WYF peptide from the final data analysis). In the absence of an internal standard, it is also difficult to assign integration boundaries in samples with low or no signal. What best practice can the authors envision to overcome these many challenges; or is the use of a labeled internal standard inevitably the next step in assay development?

13. Other than the chromogranin a spike-in, which shows considerable variability, have the authors taken any other steps to control for batch effects? Can they show the chromogranin a response as a function of batch and use the data as a measure of intra- and inter-batch variability?

14. The authors rationalize the “qualitative” approach based on the variability of the heterogeneity of the sample type. Nonetheless, it seems imperative that there is some measure of protein recovery, either using a traditional protein assay or mass spectrometry. It would be important to know what percentage of PCR and MS negative samples have low protein yield and are thus

potential false negatives. This is perhaps a question that can be more easily addressed by a MS approach. Can the assay be modified to include a measure of recovery of host protein abundance (e.g. albumin)? It is possible to re-analyze a subset of samples to estimate the relative protein recovery based on a host protein?

15. It would be helpful to clearly describe, in a table, the validation tests that have been performed to date and the results (quantitative or otherwise) of these studies.

16. Can the authors discuss the relative merits of this qualitative approach versus a quantitative assay which would employ an external calibration curve? Qualitative assays are certainly valid, for example in the detection of expressed variants (PMID: 32234366). However, would it not be more suitable to measure the LOD/LOQ based on a calibration curve (using NCAP spiked into matrix pre-digestion) rather than S/N? How would you translate this qualitative assay across labs, especially when any standards (isotope-labeled or otherwise) are not built into the assay?

Reviewer #2 (Remarks to the Author):

The authors described in the manuscript entitled "Fast and low-cost detection of SARS-CoV-2 peptides by tandem mass spectrometry in clinical samples" the development of an automated high-throughput targeted LC/MS approach for the detection of SARS-CoV-2 based on the detection/quantification virus-derived proteins. The assay was validated on more than 550 clinical nasopharyngeal/oropharyngeal swab samples and they compared their findings with the current gold standard test, i.e. RT-PCR, showing very promising AUROCs. The manuscript is mainly focused on the optimization of the sample processing (automated SP3 protocol using a liquid handling robot) and the sample acquisition (turbulent flow chromatography connected to an SRM-based MS instrument). Despite the enormous amount of work presented, and despite its relevance in the context of the worldwide COVID-19 emergency, and despite the high quality of the write-up, several issues have to be addressed before the manuscript can be accepted in this peer-review journal.

Major comments (arbitrary order):

1. Since the authors developed a very short LC-SRM targeted method, the reproducibility of the retention time is critical and a figure showing the stability of the retention time across all the

samples would be very useful and would add some validation to their analytical method especially since they mentioned the presence of carryover.

2. The main thrust of the manuscript is not clear: from reading the manuscript, it seems that the main focus was the LC method development/implementation. However, a dearly needed figure is missing (not everybody knows the concept of turbulent flow LC and how an online multiplexing is achieved). Instead, a figure for the sample processing is provided. This seems to be a confusing disconnect, which should be addressed.

3. It would be helpful if e.g. a PCA plot of all the samples was shown to check for possible batch effects due to the sample processing (96 samples per batch). Furthermore, details about the cohort are missing: we know that 563 samples were analyzed, but no information about the true positives and true negatives (based on the RT-PCR) is provided.

4. The choice of their peptides selected for SRM is not completely clear and should be elaborated on as the selected peptides seem to be amongst the less intense peptides based on their PRM results (see Supplementary Figure 2). Furthermore, the role of the WYF (WYFYLLGTGPEAGLPYGANK) peptide is not clear. Why showing it in Figure 2, if it is not used for anything (per Figure 4). Some streamlining of the entire workflow and decision trees might be helpful to better appreciate the decision making.

5. The authors might want to reconsider the order of the figures to make the manuscript more comprehensible: for instance, Figure 2 and 3 should be swapped, and Figure 4 should be moved to the supplementary data while Supplementary Figure 4 should go into the main text.

6. Re Table 4: The provided 95% CI for the combination of the parameters does not seem to be correct. Based on SPEC, SENS, PPV and NPV, the combination seems to be superior by quite a bit, but this is not reflected by the 95% PI – is it possible that there is a copy-paste error with the previous line?

7. Speaking of the combination: The sensitivity and specificity are greatly improved with a combination of parameters based on the three peptides; however very little information is given about this combination. How was this produced? What were the parameters? How does the ROC curve look of this combination against the three peptides? Is this combination of parameters purely a training set or was the data split up between training and test?

8. With a viral infection the presence of these specific SARS-COV-2 proteins is in a way binary. They are either there or not. Of course, the MS could possibly miss some of the SARS-COV-2 peptides, and thus some type II errors are inevitable. However, uninfected patients should have no peptides and therefore the number of false positives (type I error) should be low. When looking at Table 4, the specificity numbers of the three peptides do tell a different story. This is not well addressed in the discussion. There is some explanation about the carryover but even if this would be the cause I would be cautious with the sensitivity results of the provided method.

9. Along the lines of the previous issue: a more detailed discussion about the apparent systemic deviation of the LC/MS assay vs. the viral load in the lower viral load range should be discussed in more detail. Furthermore, it would be helpful to clearly indicate whether all the dots in Figure 4 represent a positive case (i.e. the negative cases are absent) or whether somewhere a cut-off is applied to define positives vs. negatives.

Minor comments:

1. The authors should mention the total protein quantity in the samples as well as the peptide quantity injected onto the LC column.

2. A kinetic of the digestion time might be helpful to optimize the sample processing and potentially increase the throughput.

3. Please provide the definitions for the DGI and IGM abbreviation also in the main text and not only in the figure legend.

RESPONSE TO REVIEWERS

REVIEWER #1

Bottlenecks in the availability and other challenges in viral testing for SARS-Cov-2 have raised the idea that mass spectrometry might complement RT-PCR-based testing. To my knowledge, the authors of this manuscript present the first targeted MS assay for SARS-Cov-2 to be deployed on a large number of clinical specimens; and the cross-comparison with RT-PCR results is important. Although they do acknowledge the “qualitative” nature of the assay, there are significant limitations of the “Tier 3” approach (as defined by Carr et al. MCP 2014). Importantly, the assay lacks trypsin-cleavable stable isotope-labeled internal standards for the NCAP protein that are needed to achieve the highest confidence and assay precision and to control for matrix effects on chromatography and digestion. These and other criteria for a peptide-based clinical diagnostic have been defined in Carr et al. and in Grant and Hoofnagle (Clin Chem 2015), and elsewhere, and the authors should more clearly state the significant limitations of this work in the context of these best practices. Nonetheless, this manuscript represents an important step forward in the development of a MS-based assay for SARS-Cov-2, particularly in the demonstration that viral proteins can be detected in respiratory sample types that are currently used for PCR-based assays. While the demonstration of throughput is important, it should not outweigh concerns about assay sensitivity and specificity. The authors should address these specific comments:

We thank Reviewer 1 for all comments, critics, and suggestions. All points indicated are addressed in this version. The most significant addition we did was the incorporation of the Protein Standard Absolute Quantification (PSAQ) approach, which was motivated by related questions from reviewers. We expressed and purified recombinant unlabeled and ¹⁵N-labelled NCAP protein, which now is used for isotope dilution to address limitations related to a Tier 3 approach. Therefore, the stable isotope internal standard co-elutes with the targeted peptides, is fragmented to yield the corresponding, mass-shifted peptide backbone fragment ions, have identical relative abundances of the fragment ions and compensate for ion suppression and poor spray stability. Full-length isotope-labeled NCAP is subject to all sample preparation steps. Now, for the Tier 1 assay, a calibration curve was prepared using recombinant NCAP quantified by amino acid analysis. A new set of clinical specimens were analyzed using the quantitative assay and an additional validation is provided that attests reproducibility, accuracy by comparison with real time RT-PCR, the detection and quantification limits, and linearity using different matrices. The reference for best practices in clinical proteomics and other methods related to bioanalytical validation have been added to the text (page 2, line 43; page 10, line 20 and 33; page 16, line 29 and 44; page 16, line 29). Despite the fact that the current strategy still aims at qualitative detection, the additional quantitative information is useful since it allows comparison between Tiers 1 and 3 and will further help to set up other tests directed to the antigen.

1. The claim of assay cost does not belong in the title or abstract of this paper. There is no evidence presented that a MS assay can be performed at lower cost than RT-PCR, which would require a detailed accounting of capital equipment and operational

(upkeep, personnel) costs, in addition to reagent costs. An emphasis in the title of the “proof-of-concept” nature of this work would be more appropriate.

The title was modified to “Fast detection of SARS-CoV-2 peptides by tandem mass spectrometry in clinical samples: proof-of-concept” and low cost was removed from the abstract.

2. The abstract mentions a shortage of reagents and instruments as a limitation to initial large-scale screening. These problems would likely have occurred for an MS test as well, and the lack of availability of sampling devices (e.g. swabs) and PPE for health care workers would be no different for these two technologies. There should be no debate that translation of an assay across laboratories would require the production of large amounts of reference material (e.g. recombinant NCAP protein) and stable isotope-labeled standards. Both of these would take considerable time and resources. These limitations will clearly need to be acknowledged in the manuscript.

We removed the following sentence from the abstract: “Real-time reverse-transcription PCR (real-time RT-PCR) is the gold standard test for virus detection but the soaring demand for this test resulted in a shortage of reagents and instruments, severely limiting its applicability to large-scale screening.” Also, we added a paragraph in the conclusion where we acknowledge these limitations (page 11, lines 16-22). Besides, the sub-heading Data Availability, which now is called Data and Material Availability (in the very end of the manuscript) includes the following statement: “Calibration curve aliquots, vectors, and/or transformed bacterial clones for isotope standard production and method standardization may be provided under request”.

3. The abstract should clearly state the qualitative nature of the assay and that it is a “Tier 3” assay based on the guidelines of Carr et al. (MCP 2014). The description of the “¹⁵N-labeled global standard” is somewhat misleading since it can be interpreted as a NCAP standard. Rather, this should be replaced with “surrogate standard” which more accurately describes this chromogranin a standard.

The qualitative assay is now properly assigned as Tier 3 assay. A Tier 1 assay based on the Protein Standard Absolute Quantification approach described by Brun and colleagues (PMID: 17848587) was validated following what is described in references for good practices in clinical proteomics. Tier 1 and 3 were compared. The term surrogate standard is now used for the chromogranin A across the text (page 5, lines 22 and 35; page 6, line 2; page 7, lines 2 and 3; page 8, line 7; page 13, line 31; page 16, lines 9 and 11; page 23, line 6).

4. If word limits permit, it would be informative, in the abstract, to specify the respiratory fluid types analyzed (nasopharyngeal and oropharyngeal swabs, and nasal wash) and the n of each that was included in the validation study. Abbreviations, including UPLC and MS/MS should be avoided if possible.

We worked with clinical respiratory tract samples that combined nasopharyngeal and oropharyngeal swabs. Less than 4% of samples analyzed were washes and we no longer mention them in the manuscript. Also, in the new quantitative validation set, all specimens are a combination of nasopharyngeal and oropharyngeal swabs and this information has been added to the abstract. UPLC was replaced by ultra-

performance liquid chromatography and MS/MS by tandem mass spectrometry in the abstract. Other adjustments were necessary to keep within the abstract word limit.

5. The introduction should include peer-reviewed citations or for the shortage of SARS-Cov-2 tests (rather than “lay media” like the New Yorker), and the discussion of serological testing should have appropriate citations. The authors should acknowledge (and provide citations for) significant prior art with respect to clinical peptide-based assays that are either in use or under development, including those for thyroglobulin and apolipoproteins. In this context, the authors should also cite the guidelines for best practices referenced above.

Lay media references were removed from the introduction and replaced by a Nature reference (page 2, line 9). References for targeted proteomics assays for thyroglobulin, apolipoproteins, and glycosylated hemoglobin are now cited in the text including quantitative proteomics assays based on PSAQ (page 2, lines 24-26). Guidelines for best practices are cited throughout the text (page 2, line 43; page 10, line 20 and 33; page 16, line 29 and 44; page 16, line 29).

6. The viral assay for HMPV used a 60 min assay (w/30 min gradient), not a 90 min assay as described in the introduction. The authors should also note that similar approaches have been extended to the identification of bacterial pathogens in respiratory tract samples (PMID: 31941798; another example of a “proof-of-concept” that did not include stable isotope-labeled internal standards for analytes of interest).

A new paragraph in the introduction discussing other approaches to identify bacterial in clinical samples based on mass spectrometry was added and the references indicated (page 2, lines 29-32). Also, we corrected the information from Foster and colleagues (page 2, line 39).

7. Throughout the paper, the authors should be specific about the type(s) and numbers of respiratory tract samples used at various steps. There are mentions of nasopharyngeal swab, oropharyngeal swab and nasal wash. The methods for collection and storage of each of the sample types should be clearly defined. For example, how many and what samples were stored in virus transport media (please cite the source) versus sterile saline.

Two worksheets were uploaded to Panorama Web (in subfolders Tier 1 and Tier 3, Master datasheet) describing the storage media, real time RT-PCR result and type of experimental validation. The original reference for the virus transport media was included (page 12, line 4).

8. The availability of both DDA and PRM/SRM data for re-analysis by the scientific community is noted. Could the authors also provide data for all of the validation studies and provide metadata, specifically the clinical sample type, for all of the individual samples?

All validation data and the Master datasheet were uploaded to Panorama Web.

9. A re-analysis of the DDA data in gpmdb (<https://gpmdb.thegpm.org>) confirms the identification of SARS-Cov-2 NCAP but also shows that cysteine alkylation was ineffective. This could be due to a lack of excess IAM versus DTT (5 mM of each was used). This should be noted although it does not affect that overall results and subsequent steps used a reduction step (w/o alkylation). In addition, the .blib file in ProteomeXchange should not be the “redundant” version as this will not import into Skyline.

Thank you very much, you are totally correct. The uploaded raw data was not alkylated. Actually, only the first DDA experiment included cysteine alkylation. After optimizing the method, alkylation was removed. DDA data were reacquired to reflect the conditions of final sample preparation and we wrongly indicated iodoacetamide derivatized residue and kept this fixed modification in MaxQuant. As you indicated, it does not affect the overall results since we avoided cysteine-containing peptides. Nevertheless, we corrected this information in the text (page 13, line 9). We reuploaded MaxQuant reprocessed data and included a new non-redundant and non-alkylated .blib. The update PRIDE Project is PXD021328.

10. The potential for mutations in SARS-Cov-2 that compromise the long-term utility of the assay are a concern. In addition to the analysis presented by the authors as supplemental data, there have now been a number of reports that have documented known mutations over much larger datasets that should be referenced (e.g PMID: 32092483 and 32387564). It is worth noting that the P344S mutation in NCAP is immediately upstream of the cleavage site that generates the IGMEV peptide. Can the authors comment on whether this might affect the cleavage by trypsin? Based on your experience, can you comment in the discussion whether mass spectrometry assays would be better or worse than RT-PCR in the case of a rapidly mutating virus?

Although these mutations were not detected in Brazil up to now, we expand the genetic mutations assessment (page 4, lines 26-28; page 15, lines 3-9). Also, we added a new supplementary figure depicting the mutation frequency in the coding region of the targeted peptides based on the reference indicated (suppl. figure 4), and a new supplementary table containing all AAs substitutions (insertions and deletions are not reported) that could affect DGI and IGM detection by mass spectrometry. The table was built considering all sequences submitted to GISAID and retrieved using CoV-GLUE (cov-glue.cvr.gla.ac.uk), filtered by two or more submitted sequences (suppl. table 3).

The mutation impact, which may result in false negatives due to molecular mass shifts and trypsin sites modification is now discussed and clearly indicated as a limitation of this approach (page 11, lines 3-15). However, it is expected that mass spectrometry assays are less impacted by mutations, if compared to RT-PCR, because of the fact that the code is degenerated. Gold standard RT-PCR methods such as those indicated by CDC are targeted to two nucleoprotein regions as this targeted proteomics method also did. Yet mass spectrometry is more information-rich than RT-PCR since it collects spectral information from the analyte rather than mere fluorescence signals. The simultaneous detection of two peptides reinforces the power of this strategy.

11. The SRM data should be included as a single Skyline file rather than split into a different file for each batch. In addition, while we understand the rationale for using a hybrid (small molecule and peptide) mode in Skyline, all of the data should be imported in peptide mode with a representative spectral library. This would enable calculation of a dotp for each peptide. If normalization to a global standard (chromogranin a) cannot be performed in peptide mode, this could be performed outside of Skyline.

We reorganize, now Tier 3, data and uploaded to Panorama a single Skyline file. We uploaded separate Skyline files for Tier 1 since calibrations were performed for every batch. Now, the peptide mode was used, and a representative spectral library was included with normalization to global (Tier 3) and SIL standard (Tier 1).

12. The lack of stable isotope-labeled internal standards is a major limitation of the approach that needs to be discussed in detail. As mentioned above, clinical MS assays are expected to utilize spike-in of SIL peptide (winged, cleavable) or SIL protein prior to trypsin digestion. Despite the validation data presented, there is the potential for false-positive detection based on isobaric interferences. Without the internal standard, there is the potential for a false negative when a peptide has gone undetected because it has fallen outside of the measured retention time (this could be a rationale for excluding the WYF peptide from the final data analysis). In the absence of an internal standard, it is also difficult to assign integration boundaries in samples with low or no signal. What best practice can the authors envision to overcome these many challenges; or is the use of a labeled internal standard inevitably the next step in assay development?

We agree with the reviewer about the difficulties related to the lack of SIL. We worked several weeks uninterruptedly to incorporate a stable isotope standard, to design a Tier 1 test, and to validate it. As for Tier 3, where a surrogate protein standard was included in the first step of sample preparation, now Tier 1 is based on a ¹⁵N-labeled nucleoprotein as an isotope-labeled internal standard. A comparison between Tiers 1 and 3 is provided. WYF peptide was excluded from the analysis since it does not improve method performance. In addition to monitoring SIL peptides across the newly acquired dataset, we also included within the 2.5 min mass detection window a peptide from endogenous beta actin protein to monitor collection efficiency and reduce the probability of false-negative detection. Beta actin is also used as an internal control in RT-PCR tests.

13. Other than the chromogranin a spike-in, which shows considerable variability, have the authors taken any other steps to control for batch effects? Can they show the chromogranin a response as a function of batch and use the data as a measure of intra- and inter-batch variability?

To control for batch effects and to ensure data quality, system suitability tests and quality controls check were monitored for every batch. The system suitability process, which includes tests for peak resolution and width, retention time, signal stability, and instrument response, does the work of checking the LC-MS instrument performance before the analytical run. Quality control checks are used to evaluate the entire workflow. Positive and negative quality controls were included in the beginning and end of each run. If the control failed to yield the expected results, the

run was rejected. System suitability and quality control are now more properly described in the text (page 8, lines 33-40; page 16, line 7). We also included supplemental figures to present retention times (figure 8) and peak areas variation across five days for Tiers 1 and 3 (figures 9 and 10).

14. The authors rationalize the “qualitative” approach based on the variability of the heterogeneity of the sample type. Nonetheless, it seems imperative that there is some measure of protein recovery, either using a traditional protein assay or mass spectrometry. It would be important to know what percentage of PCR and MS negative samples have low protein yield and are thus potential false negatives. This is perhaps a question that can be more easily addressed by a MS approach. Can the assay be modified to include a measure of recovery of host protein abundance (e.g. albumin)? It is possible to re-analyze a subset of samples to estimate the relative protein recovery based on a host protein?

We thank the reviewer for the suggestion. We incorporated a host protein, beta actin, to Tier 1 as an additional layer of quality control to verify collection variability. Unfortunately, Tier 3 specimens are no longer available for analysis. Supplemental figure depicting beta actin peptide variability over several batches has been included (Suppl. figure 9) and the use of the host protein as an additional qualifier is discussed (page 8, lines 41-46 and page 9, lines 1-5).

15. It would be helpful to clearly describe, in a table, the validation tests that have been performed to date and the results (quantitative or otherwise) of these studies.

A new table summarizing all validation tests performed in Tiers 1 and 3 were included in the manuscript (Table 3).

16. Can the authors discuss the relative merits of this qualitative approach versus a quantitative assay which would employ an external calibration curve? Qualitative assays are certainly valid, for example in the detection of expressed variants (PMID: 32234366). However, would it not be more suitable to measure the LOD/LOQ based on a calibration curve (using NCAP spiked into matrix pre-digestion) rather than S/N? How would you translate this qualitative assay across labs, especially when any standards (isotope-labeled or otherwise) are not built into the assay?

We agree that our quality control samples, which would allow inter-assay comparison would be limited for an inter-laboratory assay. Our attempts to correlate the three qualifiers with RT-PCR allow a certain degree of interlaboratory comparability. We believe that with the incorporation of the Tier 1 assay and comparisons provided between Tier 1 and 3, the issues raised are answered.

Reviewer #2 (Remarks to the Author):

The authors described in the manuscript entitled “Fast and low-cost detection of SARS-CoV-2 peptides by tandem mass spectrometry in clinical samples” the development of an automated high-throughput targeted LC/MS approach for the detection of SARS-CoV-2 based on the detection/quantification virus-derived proteins. The assay was validated on more than 550 clinical nasopharyngeal/oropharyngeal swab samples and they compared their findings with the current gold standard test, i.e. RT-PCR, showing

very promising AUROCs. The manuscript is mainly focused on the optimization of the sample processing (automated SP3 protocol using a liquid handling robot) and the sample acquisition (turbulent flow chromatography connected to an SRM-based MS instrument). Despite the enormous amount of work presented, and despite its relevance in the context of the worldwide COVID-19 emergency, and despite the high quality of the write-up, several issues have to be addressed before the manuscript can be accepted in this peer-review journal.

We thank Reviewer 2 for all comments, critics, and suggestions. All of them were addressed and many led to new experiments, approaches related to data analysis, and information that was added to the manuscript or are mentioned in this letter. They definitively contributed to improve our manuscript.

Major comments (arbitrary order):

1. Since the authors developed a very short LC-SRM targeted method, the reproducibility of the retention time is critical and a figure showing the stability of the retention time across all the samples would be very useful and would add some validation to their analytical method especially since they mentioned the presence of carryover.

A new figure depicting retention time variation over five days was included in the supplementary material (Suppl. Figure 10). As questions were raised by the lack of an isotope standard, we incorporated new data acquired with protein standard absolute quantification; isotope internal retention time variation is also presented.

2. The main thrust of the manuscript is not clear: from reading the manuscript, it seems that the main focus was the LC method development/implementation. However, a dearly needed figure is missing (not everybody knows the concept of turbulent flow LC and how an online multiplexing is achieved). Instead, a figure for the sample processing is provided. This seems to be a confusing disconnect, which should be addressed.

We believe both automated sample preparation and fast acquisition enabled by multiplexed turbulent flow LC are important to achieve our final goal, which is to provide a high throughput clinical alternative to RT-PCR SARS-CoV-2 testing. We have included an illustration for turbulent flow LC and multiplexed acquisition (Figure 2).

3. It would be helpful if e.g. a PCA plot of all the samples was shown to check for possible batch effects due to the sample processing (96 samples per batch). Furthermore, details about the cohort are missing: we know that 563 samples were analyzed, but no information about the true positives and true negatives (based on the RT-PCR) is provided.

A PCA plot of all the samples was included in the supplementary material (suppl. Figure 11). The distribution is similar among batches, indicating that there was no batch effect due to sample processing.

Two worksheets are provided (uploaded to Panorama Web within Tier 1 and 3 folders) describing RT-PCR results, the storage media and validation experimental type.

4. The choice of their peptides selected for SRM is not completely clear and should be elaborated on as the selected peptides seem to be amongst the less intense peptides based on their PRM results (see Supplementary Figure 2). Furthermore, the role of the WYF (WYFYLLGTGPEAGLPYGANK) peptide is not clear. Why showing it in Figure 2, if it is not used for anything (per Figure 4). Some streamlining of the entire workflow and decision trees might be helpful to better appreciate the decision making.

We found distinct ion intensities using nano/microspray and heated electrospray. There are also differences in the column chemistries between micro chromatography and TFC/narrow bore UPLC. We believe an important contribution of our study is the balance between sensitivity and speed we propose. Peptide peak shape is very important to achieve good sensitivity in the SRM method. Therefore, peptides chosen for high-throughput SRM analysis were selected due to the balance between peak shape and signal-to-noise ratios. Additionally, the similar polarity between them allowed a short MS window acquisition of 2.5 min, improving the throughput of the SRM method. In this new version of our manuscript this is more properly detailed in Results (page 5, lines 16, 18-20) and Discussion (page 8, lines 24-25, 30-31) sections. We agree about WYF peptide, which was excluded from this version since it does not improve method performance. Thank you also for the suggestion of including a decision tree, we have included one in the supplementary material (suppl. Figure 6) detailing all data processing workflow and the decision-making process.

5. The authors might want to reconsider the order of the figures to make the manuscript more comprehensible: for instance, Figure 2 and 3 should be swapped, and Figure 4 should be moved to the supplementary data while Supplementary Figure 4 should go into the main text.

We have changed the order of the figures as suggested. We also included other figures in supplementary material based on suggestions by both reviewers.

6. Re Table 4: The provided 95% CI for the combination of the parameters does not seem to be correct. Based on SPEC, SENS, PPV and NPV, the combination seems to be superior by quite a bit, but this is not reflected by the 95% PI – is it possible that there is a copy-paste error with the previous line?

Yes, there was a copy-paste error of the previous line. The new values are shown in Supp. table 3.

7. Speaking of the combination: The sensitivity and specificity are greatly improved with a combination of parameters based on the three peptides; however very little information is given about this combination. How was this produced? What were the parameters? How does the ROC curve look of this combination against the three peptides? Is this combination of parameters purely a training set or was the data split up between training and test?

The predictive model was established using the selected parameters (qualifiers) extracted from SRM analysis (S/N IGM, S/N DGI, IGM/IS, and DGI/IS) and the combination between and among them. The receiver operating curve (ROC) was built to obtain the cut-offs for each parameter, i.e., the point of maximum accuracy;

and we also included the cut-offs calculated based on the limit of blank experiments for IGMEVTPSGTWLTYTGAIK (cut-off= 1.65) and DGIWVATEGALNTPK (cut-off = 0.83). At first, the data set was not split between training and testing. However, we recalculated it, considering 80% of the data set for training and 20% for testing. The results compared to the ones which were purely allocated in the training group did not change significantly. Below we show some combinations of the parameters to illustrate how the final classification model was established (training set, n = 432). These combinations indicate that the best predictive model (best accuracy with good sensitivity and high specificity) was the combination of S/N IGM (LOB cut-off), S/N DGI (LOB cut-off) and IGM/IS (ROC cut-off). Note that the peptide WYF (WYFYLLGTGPEAGLPYGANK) and any combination with it does not improve the method performance.

A ROC curve for the final model has been included in the manuscript (Supp. fig. 5). We have also included more detailed information in methods (page 16, lines 8-14; page 17, lines 11-12) and results (page 6, lines 14-31) sections in regard to the choice of the selected parameters and how the combination was produced.

Rule	Training Set				Agreement	Agreement (95% C.I.)	
	cut-off	Sensitivity%	Specificity%	AUC			
S/N IGM	2	85.7	85.4	0.9142	0.89 to 0.94	85.6	
S/N DGI	1.14	86.9	77.3	0.8926	0.86 to 0.92	82.8	
S/N WYF	0.82	88.8	36.2	0.6555	0.60 to 0.70	66.5	
IGM/IS	0.04	92	69.2	0.8926	0.86 to 0.92	82.3	
DGI/IS	0.16	68.1	86.5	0.8387	0.80 to 0.88	75.9	
WYF/IS	0.04	63.7	79.5	0.7786	0.74 to 0.82	70.4	
S/N IGM LOB	1.65	88.4	76.8	0.9142	0.89 to 0.94	83.5	
S/N DGI LOB	0.85	89.6	64.9	0.8926	0.86 to 0.92	79.1	
ROC rule IGM and DGI		81.3	94.1		-	86.7	0.83 to 0.89
ROC rule IGM and IGM/IS		83.3	93		-	87.4	0.84 to 0.90
ROC rule DGI and IGM/IS		82.1	94.1		-	87.2	0.84 to 0.90
ROC rule IGM and WYF		78.5	90.3		-	83.5	0.80 to 0.87
ROC rule DGI and WYF		80.5	87.6		-	83.5	0.80 to 0.87
ROC rule IGM, DGI and WYF		76.5	96.8		-	85.1	0.81 to 0.88
LOB rule IGM and DGI		84.1	90.3		-	86.7	0.83 to 0.90
LOB rule IGM and IGM/IS		85.7	88.1		-	86.7	0.83 to 0.90
LOB rule DGI and IGM/IS		84.5	93		-	88.1	0.85 to 0.91
LOB rule IGM, DGI and IGM/IS		82.1	96.2		-	88.1	0.85 to 0.91
ROC rule IGM, DGI and IGM/IS		79.7	96.8		-	86.9	0.87 to 0.90
ROC rule IGM, DGI , WYF and IGM/IS		74.9	98.4		-	84.9	0.81 to 0.88
LOB rule IGM, DGI , IGM/IS e DGI/IS		65.3	98.9		-	79.6	0.76 to 0.83

8. With a viral infection the presence of these specific SARS-COV-2 proteins is in a way binary. They are either there or not. Of course, the MS could possibly miss some of the SARS-COV-2 peptides, and thus some type II errors are inevitable. However, uninfected patients should have no peptides and therefore the number of false positives (type I error) should be low. When looking at Table 4, the specificity numbers of the three peptides do tell a different story. This is not well addressed in the discussion. There is some explanation about the carryover but even if this would be the cause I would be cautious with the sensitivity results of the provided method.

The specificity calculated for each qualifier individually used the limit of blank as cut-offs. Since the cut-offs were lower than the ones calculated at the maximum accuracy from the ROC curve (for S/N IGM 1.65 versus 2.0 and S/N DGI 0.83 versus 1.14), the probability of a type I error increased. To overcome the drop in specificity, combining the qualifiers showed to improve the specificity without compromising

the sensitivity. Therefore, the reason for using the two peptides in the classification algorithm is explained. A rule was established to monitor carryover. If the occurrence of potential carryover is detected, the sample is reinjected, preceded with injections of blanks, to avoid false positive results.

9. Along the lines of the previous issue: a more detailed discussion about the apparent systemic deviation of the LC/MS assay vs. the viral load in the lower viral load range should be discussed in more detailed. Furthermore, it would be helpful to clearly indicate whether all the dots in Figure 4 represent a positive case (i.e. the negative cases are absent) or whether somewhere a cut-off is applied to define positives vs. negatives.

All the dots in Figure 4 (now supp. fig. 14) represent positive cases previously analyzed by real-time RT-PCR. The correlations between Tier 3 qualifiers and viral load are modest (Pearson's coefficients ranging from 0.6 to 0.76) and it would be difficult to indicate a trend in systematic or random deviation. There are many factors associated with variability in both methods, including differences in molecular stabilities, the dynamic of synthesis and degradation of the virus genomic RNA and protein during the course of infection and the multistep nature of both methods. In the low viral loads, both methods would present higher variabilities since it reaches the limit of detection (LoD) when variance is higher. With the inclusion of the Tier 1 assay we observed better correlations for both peptides ($r = 0.78$ and $r = 0.85$, DGIWVATEGALNTPK and IGMEVTPSGTWLTYTGAIK, respectively), since the comparison between the number of proteins and viral load is more straightforward. Our goal with these figures were to establish some correlation between the two methods, although we are aware of the previously described limitations.

Minor comments:

1. The authors should mention the total protein quantity in the samples as well as the peptide quantity injected onto the LC column.

We calculated the median amount of 10.1 micrograms of injected digested protein mixture onto the LC column from a batch of 96 samples (lower quartile = 7.9 and upper quartile =13.1) using UV_{280nm} microquantification.

2. A kinetic of the digestion time might be helpful to optimize the sample processing and potentially increase the throughput.

To optimize our method, we evaluated trypsin digestion efficiency overtime to increase method throughput and sensibility. A pool of positive samples previously analyzed by RT-PCR was processed by the SP3 modified method and digested by trypsin (optimized trypsin concentration of 66 µg/mL). Each time point was evaluated in triplicate and the results from LC-MS analysis were imported into the Skyline. The areas of NCAP peptides were plotted versus incubation time (figure below). Based on this experimental data, we identified two hours of trypsin digestion as adequate (chart below).

3. Please provide the definitions for the DGI and IGM abbreviation also in the main text and not only in the figure legend.

We replaced all abbreviations used throughout manuscript for peptides DGI and IGM with their whole sequence (DGIIVVATEGALNTPK and IGMEVTPSGTWLTYTGAIK).

REVIEWERS' COMMENTS

Reviewer #1 (Remarks to the Author):

The revised manuscript is substantially improved, in particular, with the use of a stable isotope-labeled protein standard, external calibration curves, and measurements of quantitative figures of merit. One remaining criticism is that the negative/positive samples were separately grouped in the Tier 1 validation. Instead, it would have been preferable to randomize all of the samples. This would have better modeled a cohort of subjects with unknown infection status and better mitigated any batch effects.

Response to reviewer comments

Reviewer #1 (Remarks to the Author):

The revised manuscript is substantially improved, in particular, with the use of a stable isotope-labeled protein standard, external calibration curves, and measurements of quantitative figures of merit. One remaining criticism is that the negative/positive samples were separately grouped in the Tier 1 validation. Instead, it would have been preferable to randomize all of the samples. This would have better modeled a cohort of subjects with unknown infection status and better mitigated any batch effects.

Thank you once again for your valuable comments. We included a new dataset, now fully randomized using an R script. The new dataset (dataset_tier1_randomized_by_R) was upload to Panorama Web as well. As you can verify below, we found no difference in terms of accuracy with or without randomization. The new results were combined with the initial 315 samples and now we present a Tier 1 accuracy validation with 445 samples.

	Accuracy (95% CI)	Sensitivity (95% CI)	Specificity (95% CI)
Tier1- initial (n=315)	86.0% (81.8% - 89.4%)	78.0% (71.5% - 83.4%)	97.0% (92.5% - 98.8%)
Tier 1- fully randomized (n=130)	90.0% (83.6%-94.1%)	78.7% (65.1% - 88.0%)	96.4% (89.9% - 98.8%)
Tier 1 (n=445)	87.2% (83.8%-90.0%)	78.2% (72.4% - 83.0%)	96.8% (93.5% - 98.4%)